# Mesomelic dysplasias associated with the *HOXD* locus are caused by regulatory reallocations

Christopher Chase Bolt [1✉], Lucille Lopez-Delisle [1,4], Bénédicte Mascrez [2,4] & Denis Duboule [1,2,3✉]

Human families with chromosomal rearrangements at 2q31, where the human *HOXD* locus maps, display mesomelic dysplasia, a severe shortening and bending of the limb. In mice, the dominant *Ulnaless* inversion of the *HoxD* cluster produces a similar phenotype suggesting the same origin for these malformations in humans and mice. Here we engineer 1 Mb inversion including the *HoxD* gene cluster, which positioned *Hoxd13* close to proximal limb enhancers. Using this model, we show that these enhancers contact and activate *Hoxd13* in proximal cells, inducing the formation of mesomelic dysplasia. We show that a secondary *Hoxd13* null mutation in-*cis* with the inversion completely rescues the alterations, demonstrating that ectopic HOXD13 is directly responsible for this bone anomaly. Single-cell expression analysis and evaluation of HOXD13 binding sites suggests that the phenotype arises primarily by acting through genes normally controlled by HOXD13 in distal limb cells. Altogether, these results provide a conceptual and mechanistic framework to understand and unify the molecular origins of human mesomelic dysplasia associated with 2q31.

[1] School of Life Sciences, Ecole Polytechnique Fédérale de Lausanne (EPFL), Lausanne, Switzerland. [2] Department of Genetics and Evolution, University of Geneva, Geneva, Switzerland. [3] Collège de France, Paris, France. [4]These authors contributed equally: Lucille Lopez-Delisle, Bénédicte Mascrez. ✉email: christopher.bolt@epfl.ch; denis.duboule@epfl.ch

Several human families displaying shortened and bent forearm bones have been reported with large chromosomal rearrangements in the q31 band of chromosome 2, a region containing the *HOXD* gene cluster[1–4]. Although they are correlated, the potential involvement of *HOX* genes in causing these limb dysmorphias (mesomelic dysplasias (MDs)) has never been confirmed, despite various studies in mice indicating that some *Hox* mutations can reproduce this condition. For example, the loss-of-function of either *Hoxd11* or *Hoxa11* individually produced mild phenotypes in the forelimbs[5,6], but when these paralogous null alleles were combined, a severe MD affecting both the radius and ulna appeared, resembling human arms with 2q31 alterations[7–9]. However, none of the human families evaluated by genomic analyses showed any mutations affecting *HOX* gene bodies, suggesting that the limb malformations were likely to result from mutations interfering with the highly coordinated regulation of *HOXD* gene transcription during early limb development[10,11].

A strong MD was also reported in mice carrying the X-ray-induced mutation *Ulnaless*, an inversion mapping to the murine *HoxD* locus[12,13]. Evaluation of *Hox* transcripts in the limbs of *Ulnaless* mutant embryos revealed the ectopic presence of *Hoxd13* transcripts in the presumptive cellular domain for the radius and ulna, but also gave conflicting results on the down-regulation of both *Hoxa11* or *Hoxd11* (refs. [14,15]). Because *Hoxd13* is normally transcribed only in the most distal cells of the developing limb buds where digits are formed[16], the possibility that mesomelic dysplasias in both human and mice are caused by a deleterious *Hoxd13* gain-of-function in the proximal domain, where long bones of the forearm normally develop, was put forward[15]. This hypothesis was supported by the dominant nature of these malformations in both the human conditions[17] and the mouse *Ulnaless* mutant[12], the latter being mostly homozygous lethal[14,15].

Extensive chromosome engineering at the murine *HoxD* locus has shed light on the complex regulation of these genes during limb development. The gene cluster is flanked by two ca. 1 Mb regulatory landscapes. Centromeric to the cluster (on the side of *Hoxd13*), a range of digit-specific enhancers regulate the transcription of *Hoxd13* to *Hoxd10* in the most distal cells of the growing limb bud[18] (Fig. 1a, C-DOM). On the other side of the gene cluster, a series of proximal limb enhancers activate *Hoxd9* to *Hoxd11* in developing forearm cells[11] (Fig. 1a, T-DOM). This bimodal type of regulation is made possible by the presence of an insulation boundary localized between *Hoxd11* and *Hoxd12* which is established by several bound CTCF proteins[19]. Under normal conditions, this strong insulation boundary prevents the activation of *Hoxd13* in forearm cells by proximal limb enhancers. In the *Ulnaless* allele, the *HoxD* cluster is inverted[13] (see Fig. 1a) and as a consequence, *Hoxd13* is brought into the vicinity of known forearm enhancers, putatively explaining its ectopic activation in proximal limb cells. Since the semi-dominant gain of *Hoxd13* expression coincides with a phenotype that mimics the combined loss of both *Hoxd11* and *Hoxa11* in the proximal limb, it was proposed that the presence of the HOXD13 protein would either directly repress the transcription of *Hox11* genes[14] or inhibit the function of group 11 HOX proteins through a dominant-negative mechanism referred to as "posterior prevalence"[20].

The necessity to prevent expression of *Hoxd13* in proximal limb bud cells was further documented by forcing its expression in the whole limb bud using a transgenic paradigm. While early and strong expression of the transgene completely ablated limb formation proximal to the hands and feet[21–23], another chromosomal rearrangement at the *HoxD* locus showed that a late and weak gain of *Hoxd13* transcription in the proximal limb was enough to shorten the length of the radius and ulna[24]. Altogether,

this evidence suggested that the ectopic gain of HOXD13 in presumptive forearm cells is the cause of mesomelic dysplasia. However, several key questions remained to be answered to turn this hypothesis into an explanation. For instance, how is *Hoxd13* transcription gained in proximal limb cells? Is the gained HOXD13 protein really the cause of the observed alterations and if yes, does ectopic HOXD13 produce these alterations by directly down-regulating *Hox11* transcription or does HOXD13 interfere with HOX11 protein activity in a dominant-negative manner?

In this work, we address these questions by using a chromosomal inversion in mice similar to the *Ulnaless* rearrangement, yet with slightly different breakpoints leading to a milder gain of *Hoxd13* expression and accompanying hypomorph mutant phenotype. We induce a secondary mutation in *cis* with the inversion, demonstrating that the gain of HOXD13 expression is indeed the sole reason for the mesomelic dysplasia phenotype. We also show that this gain is caused by the abnormal genomic proximity between this gene and native proximal limb enhancers. Finally, single-cell RNA-seq and protein binding analyses suggest that the deleterious effect of HOXD13 in proximal cells is mediated partly by its binding to sites normally occupied by HOX11 proteins, together with a partial reduction in the transcription of *Hoxa11*. These results allow us to present an inclusive molecular explanation for all cases of human 2q31 mesomelic dysplasias reported thus far.

## Results

**A mouse model for limb mesomelic dysplasia.** Several human mesomelic dysplasias have been associated with the *HOXD* locus[2,4,25–27] (Fig. 1a). While the *HOXD* genes themselves are not affected in these conditions, the physical relationship with the flanking regulatory regions are modified, suggesting a potential impact of chromosomal rearrangements upon the long-range regulation of these genes during early limb development[2,11,28]. The murine *Ulnaless* X-ray-induced inversion on the *HoxD* cluster is an excellent proxy to study mesomelic dysplasia. However, the severity of its effects and the early homozygous lethality, perhaps due to a breakpoint in the *Lnpk* gene (Fig. 1a, *Ulnaless*), made the use of these mice difficult for further analyses and genome editing. We circumvented these problems by using a new *HoxD* inversion (*HoxD^{inv2}* mm10 chr2: 74477755–75441001), which was engineered with a 5′ breakpoint within C-DOM, just downstream of the *Lnpk* gene, whereas the 3′ breakpoint was positioned telomeric to the gene cluster within the proximal limb regulatory domain (Fig. 1a, *inv2*). As a consequence, the *Lnpk* gene remained intact and most proximal limb enhancers were inverted along with the gene cluster (Fig. 1a, bottom panel). Since these latter enhancers were likely responsible for the strong gain of *Hoxd13* expression in the *Ulnaless* inversion, this new inversion was expected to produce a weaker phenotype and viable homozygous specimen, allowing us to carry out the necessary analyses.

This inversion was produced by the STRING approach[29] and mice were born at a Mendelian ratio, without any detectable limb anomaly in the heterozygous condition, unlike the *Ulnaless* allele. However, F2 mice homozygous for the *HoxD^{inv2}* (hereafter *inv2*) inversion displayed a clear abnormal morphology of their forelimbs, which was accompanied by a detectable problem in walking. This abnormal phenotype, reminiscent of a mild limb mesomelic dysplasia was fully penetrant. The analysis of skeletal preparations revealed that the radius and ulna were ill-formed, shortened, bent towards the posterior aspect and rotated approximately 90° along their length with respect to the position of the humerus. These combined alterations led to the observed abnormal angle between the hands and the forearm (Fig. 1b and Supplementary Video 1). This phenotype, only observed in *inv2*

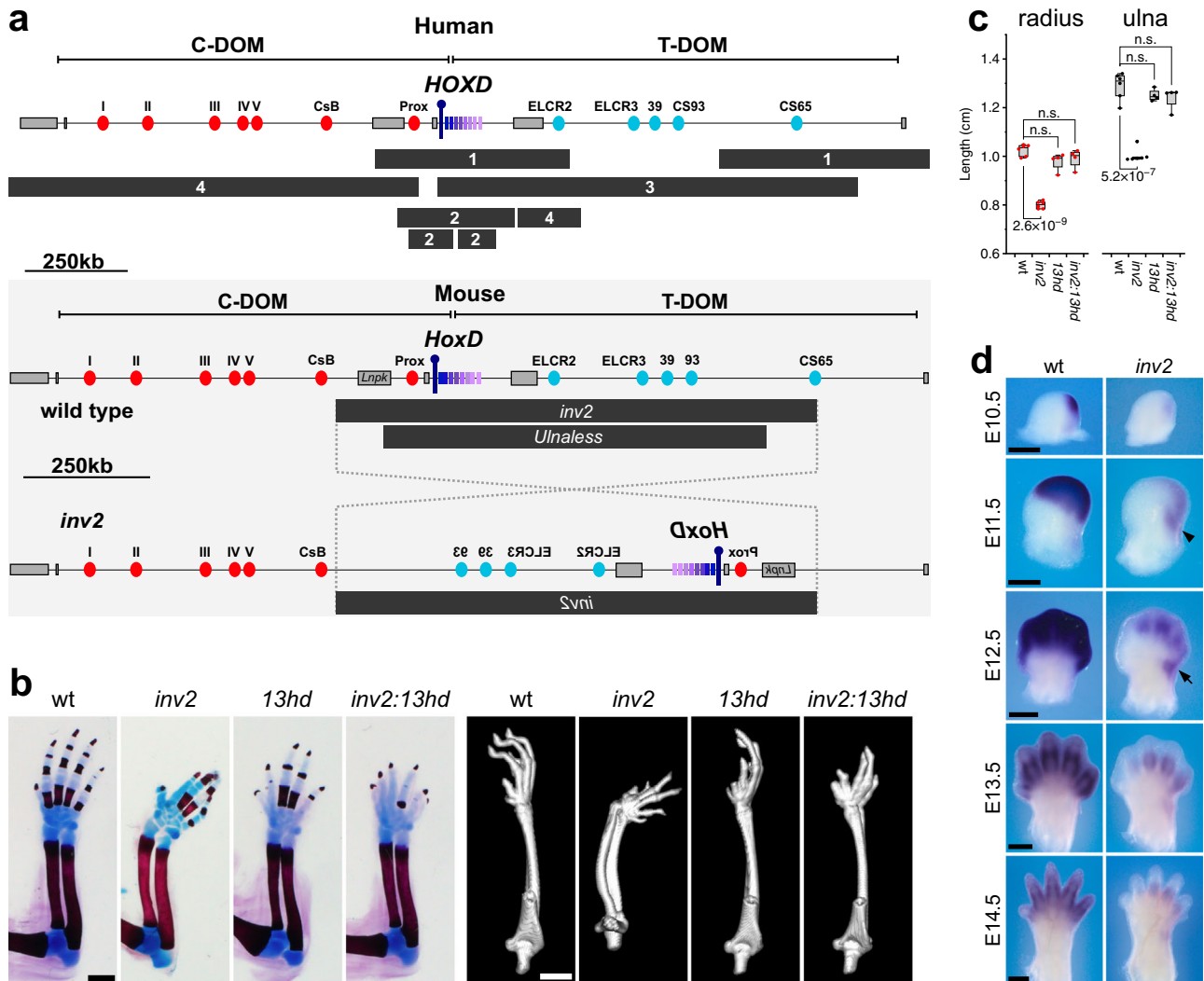

**Fig. 1 Inversion of the *Hoxd* gene cluster induces limb mesomelic dysplasia. a** The top panel shows a map of the human *HOXD* locus with the positions of mapped chromosomal rearrangements (black bars) leading to mesomelic dysplasia. The numbers inside the black bars refer to the original references: (1) Kantaputra et al. (2010)[1], (2) Le Caignec et al. (2019)[2], (3) Cho et al. (2010)[3], (4) Peron et al. (2018)[4]. The panels below (gray box) depict the wild type mouse locus (top) and the structure of the *Hoxd^inv2* (*inv2*) inverted allele. In both murine and human loci, the proximal (blue ovals) and distal (red ovals) limb enhancers are indicated, located on either side of the cluster, within the T-DOM and C-DOM TADs, respectively. *Hoxd* genes are indicated by rectangles with shades of magenta, with a purple pin to indicate the position of *Hoxd13*. The location of the mouse inversion allele *Ulnaless* is indicated below the *inv2* allele. **b** Newborn skeletal stains (left, scale is 1 mm) and adult micro-CT scans (right, scale is 2.5 mm) showing mesomelic dysplasia associated with the *inv2* allele. Inactivation of *Hoxd13* in *cis* (*inv2:13hd*) completely rescues the alteration. With the exception of the wild types, the *inv2*, *13hd*, and *inv2:13hd* mutants all show digit alterations due to significant reduction of *Hoxd13* transcripts in the distal limb (see also Supplementary Fig. 1 and Supplementary Video 1). **c** Quantification of bone length based on CT scans showing that *inv2* radius and ulna are significantly shorter than wild type control bones. The bones of *13hd* control mice do not show proximal limb defects. *inv2:13hd* radius and ulna bones are not different in length to control. Box plots are interquartile range and whiskers indicate maximum and minimum values. *P* values were determined by two-tailed Welch's unequal variances *t*-test (see Source Data File for additional details, $n$ = number of limbs measured: wt $n = 6$, *inv2* $n = 6$, *13hd* $n = 4$, *inv2:13hd* $n = 4$), ns not significant. **d** Timecourse hybridizations for *Hoxd13* mRNAs showing both the decrease of transcripts in distal cells and the ectopic proximal expression domain (arrow), most visible at E11.5 and E12.5. Scale bars are 0.5 mm. All samples are homozygous for the indicated genotype.

mice, was confirmed and further assessed after micro-CT scans of several mutant and control skeletons (Fig. 1b, right panels). CT scans allowed for precise measurements of bone lengths and revealed a significant shortening (ca. 20%, $p < 1e-6$) of both radius and ulna (Fig. 1c). In addition, the digits of *inv2* mice were abnormal, showing a pattern reminiscent of a partial loss of function of *Hoxd13* (Fig. 1b and Supplementary Video 1). Therefore, the *HoxD^inv2* allele produced living homozygous mice with a light yet fully penetrant and significant limb mesomelic dysplasia.

**Ectopic *Hoxd13* transcription and phenotypic rescue through a secondary mutation.** We determined whether the *inv2* allele had expectedly induced an ectopic expression of *Hoxd13* in developing forearm cells, as for the *Ulnaless* inversion, by performing timecourse analyses of *Hoxd13* mRNA by whole-mount in situ hybridization (WISH) (Fig. 1d). During the earliest phase of *Hoxd13* expression, we observed a weak staining when compared with wild type limb buds. Shortly after, by E11.5 when the proximal and distal limb domains begin to separate, a clear *Hoxd13* signal was apparent in the posterior-distal portion of the

nascent proximal limb domain of *inv2* mutants and absent from control littermates (Fig. 1d, black arrowhead).

From E12.5 to E14.5, the ectopic domain of *Hoxd13* mRNAs continued to be detected in the posterior-distal part of the proximal limb domain separated from the distal domain by a thin strip of low-expressing cells, i.e. matching the position of the future distal end of the ulna (Fig. 1d, arrow). While this ectopic domain was fully penetrant, it was clearly weaker and smaller than in the *Ulnaless* mutant limb buds[14,15], probably due to the fact that the strong proximal limb enhancer CS65 (ref. [11]) is not present close to *Hoxd13* in the *inv2* allele, unlike in *Ulnaless*, the inversion leaving only a few putative enhancer sequences[11] at their initial positions (see below). By E13.5 the transcription of *Hoxd13* was diminished in the proximal and distal limbs of *inv2* limbs.

To demonstrate that this localized ectopic domain of *Hoxd13* mRNAs was indeed causative of the limb mesomelic dysplasia phenotype, we used a CRISPR-Cas9 approach to induce a secondary mutation in *cis* with the inverted chromosome to functionally inactivate the HOXD13 protein (Supplementary Fig. 1a). We induced a 7 bp deletion causing a frameshift mutation N-terminal to the nuclear localization signal and homeodomain of HOXD13, the latter domain being necessary for binding to the major groove of target DNA sites. Concomitantly, the same 7 bp deletion was also isolated on the wild type chromosome as a control allele (Supplementary Fig. 1b). Mutations disrupting formation of the HOXD13 homeodomain were shown to induce a loss-of-function phenotype in the distal limbs and indeed mice homozygous for this *Hoxd13^hd* allele alone displayed the well described *Hoxd13* loss-of-function phenotype in their digits[30] (Fig. 1b). While mice homozygous for this *Hoxd13^hd* mutation in *cis* with the *HoxD^inv2/inv2* inversion (*HoxD^inv2/inv2*:*Hoxd13^hd/hd*) also displayed the expected loss-of-function phenotype in their digits, the mesomelic dysplasia was completely rescued with full penetrance, leading to normal forelimbs as verified by both skeletal staining and micro-CT analyses (Fig. 1b, c and Supplementary Video 1). This result demonstrated that the gain of *Hoxd13* function in proximal cells was indeed the unique cause of limb mesomelic dysplasia.

**Topological reconfiguration of enhancer–promoter interactions after inversion.** Expression of *Hoxd* genes during limb development is controlled by two large regulatory landscapes (Fig. 1a, C-DOM and T-DOM) which also match two topologically associating domains (TADs)[31–33] that flank the gene cluster (Fig. 2a). The insulation boundary between these two TADs[19] relies upon the presence of multiple CTCF sites that split the gene cluster into two distinct parts[34]. *Hoxd13* is located centromeric to the TAD boundary (Fig. 2a, purple pin) and responds exclusively to the various digit enhancers localized in C-DOM (Fig. 2a, red ovals), which are active only in distal limb cells. In contrast, enhancers within T-DOM (Fig. 2a, blue ovals) are active and promote *Hoxd9*, *Hoxd10*, *Hoxd11*, and *Hoxd12* transcription in the proximal limb. In the *inv2* allele, the breakpoints of the 963 kb inversion lie on either side of the *HoxD* cluster, positioned within each one of the two TADs (Fig. 2a, dashed lines). In order to assess the regulatory reallocations induced by these topological modifications, we collected cells from wild type and *inv2* mutant proximal and distal forelimbs and measured DNA–DNA interaction frequencies by Capture Hi-C (CHi-C). The captured sequences were aligned to an *inv2* mutant genome reconstructed in silico (Fig. 2a, bottom profile).

In distal limb cells of E12.5 control embryos, the chromatin conformation displayed the well-characterized topology of the

*HoxD* locus containing two TADs separated by the CTCF-dependent insulation boundary present within the gene cluster[19]. In the *inv2* allele, a major redistribution of contacts was observed, which could be explained by the reorganization of the various CTCF insulation boundaries present over the entire locus[35]. We defined insulation boundaries as a concentration of occupied CTCF sites capable of producing a bidirectional boundary effect in normal limb cells[36]. Five such boundaries were identified, schematized as triangles and labeled from 1 to 5. The orientation of the sites is reflected by the orientation and color of the triangles, where sizes indicate the strength of the boundary effect (Fig. 2a, middle panel).

On the inversion allele, the *HoxD* insulation boundary (Fig. 2a, triangles 2 and 3) was inverted and moved closer to the telomeric end of T-DOM. From this new position, one boundary element (triangle 2) established contacts with the existing telomeric boundary (triangle 5) to induce formation of a new TAD (T-DOMneo). This small and dense domain contained *Hoxd13*, a single digit enhancer (Prox)[37] and a few putative proximal limb enhancer elements that remain in place since they are located beyond the telomeric inversion breakpoint (the blue portion of T-DOMneo). All other proximal limb enhancers were relocated to the other side of the locus (Fig. 2a, between boundary triangles 1 and 3) where they also formed a new TAD structure with the large portion of C-DOM that had stayed at its initial position (Fig. 2a, C-DOMneo between boundaries 1 and 3). Thus, the C-DOMneo regulatory landscape contained the majority of functional proximal and distal limb enhancers (Supplementary Fig. 2b).

To determine if the reconfiguration of the regulatory landscapes had altered the expression of other *Hoxd* genes, we evaluated their expression by WISH. In *inv2* mutants we found a minor increase in *Hoxd11* and *Hoxd12* expression in the proximal limb buds (Supplementary Fig. 2c) likely resulting from the same change in contacts affecting *Hoxd13* expression. In contrast, we observed a reduction of *Hoxd12*, *Hoxd11*, and *Hoxd10* in the distal limb compartment, certainly resulting from the increased distance between these genes and their distal limb enhancers within the C-DOMneo configuration. The fact that only the Prox distal limb enhancer remained with *Hoxd13* after inversion explained the severely reduced transcription of *Hoxd13* in distal limb cells and the associated digit phenotype.

We then looked at T-DOMneo to determine if *Hoxd13* expression in proximal cells after inversion could be linked to increased contacts with the portion of the proximal limb regulatory landscape that had not been inverted (Fig. 2b). In the inverted allele, *Hoxd13* indeed established significant interactions with a region that was labeled by H3K27 acetylation in proximal limb cells at this stage[38] and which displayed several accessible chromatin regions, as assayed by ATAC-seq (Fig. 2b)[39]. The presence of putative regulatory elements within the T-DOMneo suggested that these elements may be responsible for the ectopic expression of *Hoxd13* in the proximal limb domain.

To confirm that this particular region carried some proximal limb regulatory activity, we identified six regions showing H3K27ac signal in wild type proximal limbs, ATAC peaks in the *inv2* samples, and no apparent CTCF binding. These six putative enhancer elements, CS68 and Proximal Limb Enhancers PLE01 to PLE05 (Fig. 2b and Supplementary Table 1) were generated as a concatenated element and positioned 5′ to a *lacZ* reporter construct. All transgenic embryos showing β-gal staining were positive in the proximal limb (7 out of 7); however, the size of the expression domain varied from a small discrete patch to several larger regions extending towards the distal boundary of the proximal limb domain (Fig. 2b and Supplementary Fig. 2e).

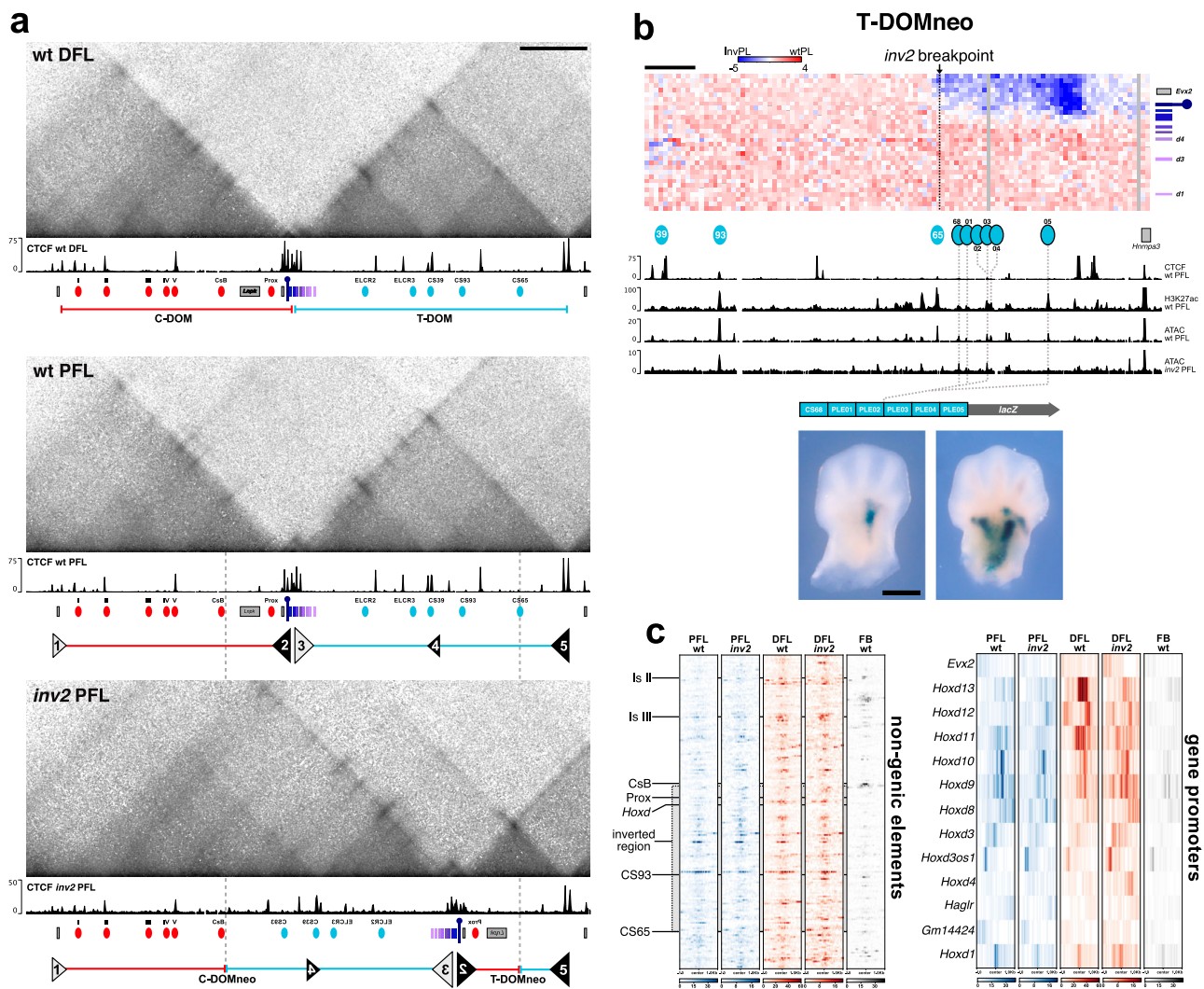

**Fig. 2 *Hoxd13* establishes new contacts with proximal limb enhancers in the *inv2* allele. a** Capture Hi-C using E12.5 wild type distal (top, DFL) and proximal (center, PFL) limb cells, as well as mutant *inv2* proximal limb cells (bottom, PFL). Bin size is 5 kb, color scale is log transformed. Wild type samples are mapped to wild type chr2 and *inv2* is mapped on a reconstructed mutant chromosome 2 (mm10). Below each heatmap are CTCF CUT&RUN tracks produced from the indicated tissues and the black and white triangles indicate the orientations of large (large triangles) or small (small triangles) groups of occupied CTCF sites. The gray dashed lines between the wild type PFL and *inv2* PFL indicate the breakpoints of the inversion. In the *inv2* allele, the observed changes in chromatin contacts matches the expectations based on groups of CTCF sites with convergent orientations. The groups are labeled from 1 to 5 to facilitate the reading of the profile after inversion. *Hoxd* genes are colored in shades of magenta and the position of *Hoxd13* is indicated with a purple pin. **b** The top panel shows the subtraction between the CHi-C contacts established by *Hoxd13* and proximal limb enhancers in wild type and in *inv2* proximal limb cells (each bin is 5 kb). Blue bins represent contacts more frequently observed in the *inv2* proximal limb cells and are concentrated in the T-DOMneo domain, starting right after the position of the breakpoint (vertical line). The mapping is shown on the wild type chromosome for clarity (chr2: 75129600–75677400). Scale bar is 55 kb. The tracks below show CTCF CUT&RUN, H3K27Ac ChIP[19], and ATAC using E12.5 wild type or *inv2* proximal forelimb cells, mapped onto wild type mm10. Previously characterized proximal limb enhancers are indicated by blue ovals below the heatmap. Putative proximal limb enhancers that gain contact with *Hoxd13* in the *inv2* allele are indicated by blue ovals with a black border. These elements (CS68 and PLE01 to 05) were identified through the H3K27Ac and ATAC profiles in wild type and *inv2* proximal forelimbs and were cloned into a single *lacZ* reporter transgenic construct. Representative staining patterns in forelimb buds are shown below (see Supplementary Fig. 2e). All embryos that stained (7 of 7) produced a variation of the proximal limb staining. Scale bar is 0.5 mm. **c** Comparison of ATAC-seq datasets between the profiles in control and *inv2* proximal forelimbs. The heatmaps on the left are peaks from ATAC samples mapped onto non-genic elements at the *HoxD* locus, illustrating the high similarity between samples of the same tissue, regardless of genotype. The regions corresponding to the gene bodies of *Lnpk*, *Mtx2*, and the region from *Evx2* to *Hoxd1* have been removed. The heatmaps on the right are the ATAC profiles over accessible gene promoters in the *HoxD* cluster. The PFL samples are generally similar while the DFL samples show more difference, in particular on *Hoxd13* and *Hoxd11* promoters. With the exception of the PLE *lacZ* transgenic embryos, all experiments were performed on embryonic samples homozygous for the indicated genotype.

While the domains were not always completely overlapping with the ectopic *Hoxd13* domain, these results confirmed that DNA sequences at the vicinity of *Hoxd13* in the *inv2* configuration are active in proximal limb cells, where they also significantly increased their contacts with *Hoxd13*. Noteworthy, other

sequences located telomeric to TAD boundary #5 were shown to contribute to *Hoxd* gene expression in a posterior patch of proximal cells[40], which together with the sequences reported here, may all contribute to the ectopic expression of *Hoxd13* in the inverted allele.

We then evaluated if the structural changes observed in the CHi-C may have contributed to the ectopic expression of *Hoxd13* in proximal limb cells by altering the enhancer status of the two regulatory landscapes. We used our ATAC-seq datasets to identify any changes in chromatin accessibility in proximal and distal limbs of both genotypes and a wild type forebrain as control[41]. First, we selected the ATAC-seq peaks corresponding to non-genic elements located throughout the locus outside of the gene cluster. We then evaluated them visually using heatmaps and found a high correspondence between the accessibility status and the tissue of origin (Fig. 2c, left). This was confirmed by hierarchical clustering analysis of the non-genic elements (Supplementary Fig. 2d, left panel). However, when we evaluated the accessible gene promoters in the *HoxD* cluster, there were only minor differences between the PFL samples, but noticeable differences between the DFL samples, especially on *Hoxd13* and *Hoxd11* (Fig. 2c right). We repeated the clustering analysis on these regions and observed that they no longer clustered by tissue of origin, rather both *inv2* samples clustered closely with the wild type PFL sample (Supplementary Fig. 2d), and the differences are most apparent in the genomic coverage map of the gene cluster (Supplementary Fig. 2a ATAC). These changes in promoter accessibility, in particular *Hoxd11* and *Hoxd13* in the DFL, mirror the reduction in expression of these genes observed by WISH, and support the conclusion that the inversion did not change the activity status of the regulatory landscapes. Instead, the inversion altered the relationship between the genes and their enhancers by forming a new three-dimensional structure that restricted 5′ genes from their normal C-DOM enhancers, and simultaneously introduced them to proximal limb enhancers, which they are normally insulated from.

**Excluding *Hox11* transcriptional down-regulation as the main cause of mesomelic dysplasia.** Having established the mechanism leading to the gain of expression of the *Hoxd13* gene in proximal limb cells and the fact that ectopic HOXD13 in these cells is the sole cause for the limb deformities, we addressed the potential mechanisms through which the protein may achieve its deleterious effect. The limb alterations produced by either the *Ulnaless* or the *inv2* alleles are both similar to the limb phenotypes found in mice with significant reductions in *Hoxa11* and *Hoxd11* transcription (remaining expression <50% in *Hoxa11* and <50% in *Hoxd11*)[7]. One proposed explanation is that ectopic HOXD13 may abrogate transcription of *Hoxa11* and *Hoxd11* in proximal limb cells so that the mesomelic dysplasia phenotype converges towards a combined *Hox11* loss-of-function phenotype[14]. However, another study did not observe any substantial change in *Hox11* transcription in proximal *Ulnaless* limbs[15].

We revisited these results by performing in situ hybridizations for *Hoxa11* and *Hoxd11* but imaged the embryos in a timecourse through the linear phase of color development when differences in staining are more apparent (Fig. 3a and Supplementary Fig. 3a). At E12.5, we observed that *Hoxa11* transcripts are slightly but clearly reduced in a small region of the proximal limb of *inv2* mutants corresponding to the ectopic *Hoxd13* domain (arrow in Fig. 3a and arrowhead in Supplementary Fig. 3a). Under the same conditions, we also observed a similarly small but consistent increase of *Hoxd11* transcripts at the same position (arrowhead in Fig. 3a and Supplementary Fig. 3a). Even so, this partial reduction of *Hoxa11* transcripts by *Hoxd13* is not sufficient to explain the observed phenotype, especially without an at least equivalent or greater reduction of *Hoxd11* transcripts.

Because in situ hybridizations have a poor cellular resolution and are difficult to quantify, we implemented single-cell RNA-seq to evaluate a potential correlation between ectopic HOXD13 and

a reduction in the amount of *Hox11* transcripts. We micro-dissected comparable regions including the ectopic patch of *Hoxd13* mRNAs in both *inv2* and wild type limbs (Fig. 3b, dashed quadrangle) and processed them for single-cell RNA-seq using the 10X Chromium platform with 3.1 chemistry. We sequenced 5006 cells from one wild type and two *inv2* biological replicates producing 4535 and 4315 cells, with a mean number of reads per cell between 60,000 and 80,000, and analyzed with the Seurat package[42]. Clustering analysis displayed in a two-dimensional UMAP identified one main group consisting of 11 clusters (Fig. 3c). All clusters were identified in both genotypes.

In order to separate proximal from distal cell clusters, we identified the clusters where *Hoxd13* was strongly expressed in wild type cells (clusters 3, 4, 5, 9), and did the same for *Hoxa11* (clusters 0, 1, 2, 4, 6, 7, 8, 9, 11), which was strongly associated with cells also expressing the proximal limb marker *Shox2* (Fig. 3d and Supplementary Fig. 3c). In the two replicates of *inv2* we identified three clusters of cells that were present in much greater proportions than in the wild type sample (7, 8, 11) and were localized on the portion of the UMAP associated with proximal limbs. To determine if these were new cell identities, we compared our dataset to a public dataset of E12 and E13 whole forelimbs[43], where we could clearly identify these three clusters of cells within the normal whole limb (Supplementary Fig. 3b). This strongly indicates that the absence of these clusters in our single wild type sample results from slight dissection effects, or possibly a difference in the development stage of this embryo, rather than these cells resulting from expanded cell identities in the *inv2* samples.

We then tried to visually identify clusters which (1) had an increase of *Hoxd13* in the *inv2* configuration, (2) express *Hoxa11* in the wild type, and (3) express other proximal limb markers. We identified two clusters meeting these criteria (clusters 1 and 6) (Fig. 3e and Supplementary Fig. 3c). In the UMAP, these two clusters reside along the boundary between cells that are distinguished by proximal and distal limb marker genes, yet more closely associate with cells displaying a proximal limb identity (Fig. 3c, d and Supplementary Fig. 3c, d).

In single-cell RNA-seq experiments, the proportion of zeros is highly anti-correlated with the mean expression value of the gene[44]. Accordingly, we used the proportion of zeros as a proxy for the mean expression values to evaluate if the ectopic expression of *Hoxd13* in the proximal limb clusters produced a correlational effect on the expression of *Hoxa11* and *Hoxd11*. We predicted that the expression level of *Hoxd13* is higher in cells where *Hoxd13* is detected, and vice versa for cells where *Hoxd13* is not detected. When we analyzed cells with detectable *Hoxd13* we observed that the proportion of cells with *Hoxa11* was always significantly decreased compared to those cells without *Hoxd13* (Supplementary Fig. 4a). The exception to this is cluster 6 in the wild type sample because there are very few cells where *Hoxd13* is detected. Following the same reasoning, we found a strong positive correlation between *Hoxd11* and *Hoxd13* in clusters 1 and 6 of the *inv2* sample.

However, this method can be biased when the number of UMIs is different between samples. In order to remove this bias, we used a new method, baredSC, to infer the true distribution of expression for each gene independently, by cluster and by genotype[45]. This new method can be applied to any single gene, determining the confidence interval on the distribution of expression, and a confidence interval on the fold-change between two conditions (Supplementary Fig. 4b). Using this approach, we found a 1.7–3.0-fold increase in *Hoxd13* in clusters 1 and 6, and observed a 40–55% decrease in *Hoxa11* transcripts compared to wild type cells in the same clusters, and a decrease of 15–30% for *Hoxd11*.

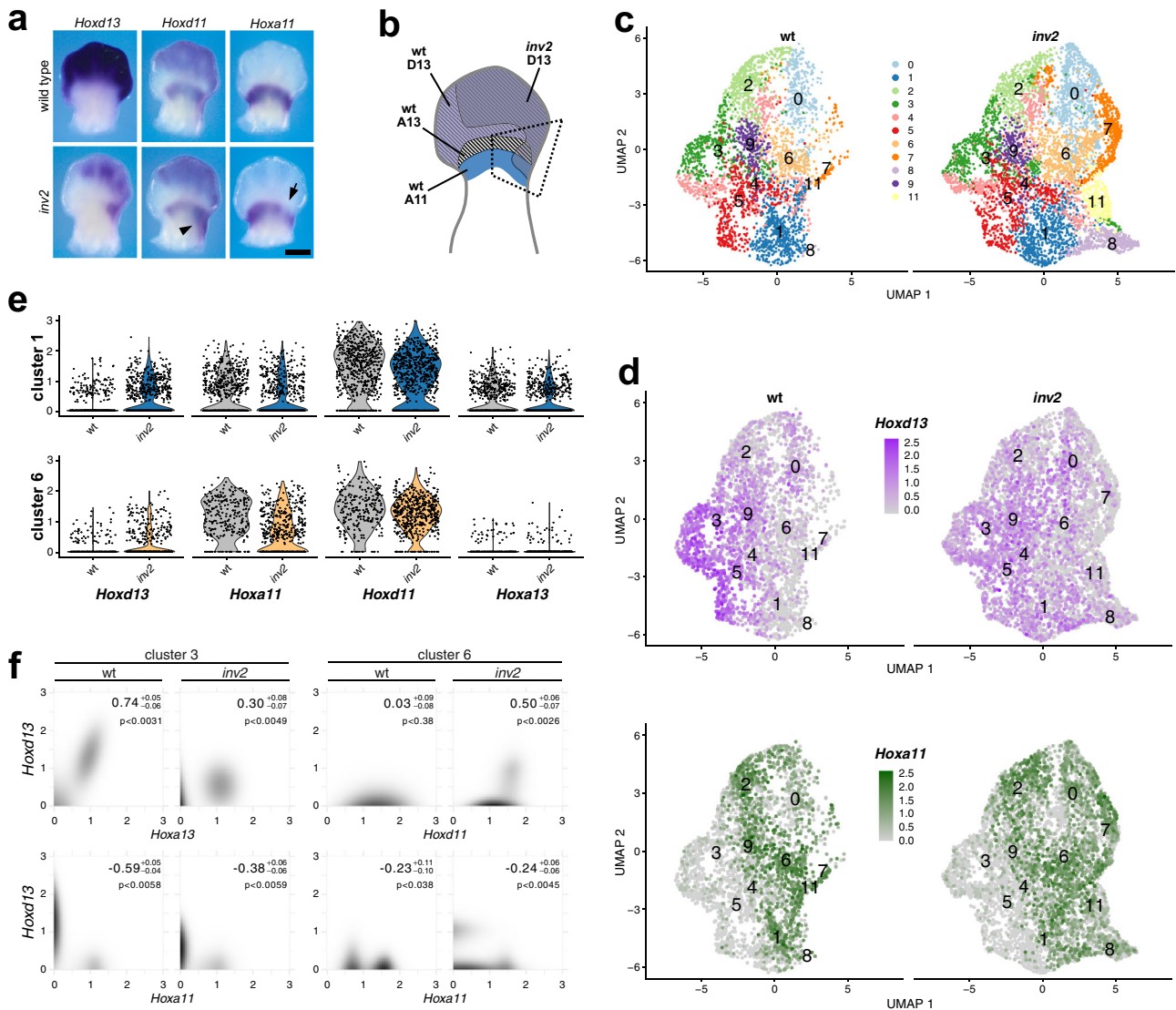

**Fig. 3 Single-cell RNA-seq analysis of the *inv2* limb ectopic *Hoxd13* domain. a** In situ hybridizations comparing the ectopic gain of *Hoxd13* in the proximal forelimb to the expression of *Hoxa11* and *Hoxd11*. A small decrease in *Hoxa11* transcripts was scored in the posterior portion of proximal limb domain (arrow), whereas *Hoxd11* staining is slightly increased in the posterior and distal portion of the proximal forelimb (arrowhead) when compared with wild type limb buds. Scale bar is 0.5 mm (see also Supplementary Fig. 3a). **b** Scheme of an E12.5 limb with the regions of gene expression for wild type (*Hoxa11* in blue, *Hoxd13* in purple, and *Hoxa13* in black and white stripes, as well as the approximate location of *Hoxd13* mRNAs in the *inv2* allele (gray). The region outlined by the black dashed quadrangle was dissected for single-cell RNA-seq analysis. **c** UMAP representation of the primary cellular clusters as determined by Seurat. All clusters were found in both genotypes, but clusters 7, 8, and 11 were present in greater proportion in the *inv2* samples than in the wild type. **d** The top panel is the UMAP representation for the main cluster with cells expressing *Hoxd13* indicated in shades of purple according to the level of expression, and *Hoxa11* in shades of green. In the wild type sample, *Hoxd13* expression was primarily limited to the distal limb clusters (3, 4, 5, and 9). With the *inv2* sample, however, the expression was reduced in distal limb clusters while it increased in the proximal limb clusters 1 and 6. **e** Clusters 1 and 6 were the only ones that expressed both *Hoxa11* and *Hoxd11*, along with a gain of *Hoxd13* in the *inv2* configuration. The violin plots represent the detected transcript expression level for each cell in that cluster. **f** Heatmaps representing the inferred distribution of cells for the different levels of expression of *Hoxd13* on the *y*-axis and *Hoxa11*, *Hoxd11*, or *Hoxa13* on the *x*-axis. The color scale is log transformed in order to see a greater range of frequency. Bins with black represent a high proportion of cells whereas bins with white indicate an absence of cells. In the right corners are indicated the confidence interval of the correlation as well as an estimation of the one-sided *p* value (probability that the correlation has the opposite sign). All samples are homozygous for the indicated genotype.

This method can be extended to evaluate the distribution of expression for two genes at the same time. First, we evaluated the distal limb control cluster 3 and found a clear anti-correlational between *Hoxd13* and *Hoxa11* (Fig. 3f, bottom left panel), matching the previous observation that *Hoxd13* represses the transcription of *Hoxa11* in the distal limb[38,46]. In the same cluster we found a positive correlation between *Hoxd13* and *Hoxa13*, which is also not surprising due to the high frequency of these

genes being expressed in the same distal limb cells[47]. Finally, we evaluated the proximal limb cluster 6 and found a clear anti-correlation between *Hoxd13* and *Hoxa11*. Of note, the expression of *Shox2* was not decreased in *inv2* cells from cluster 6, showing that the mesomelic phenotype was not induced by a secondary effect through this proximal limb gene, which in human is the causative agent of other types of limb reductions associated with short-stature syndromes[48–50].

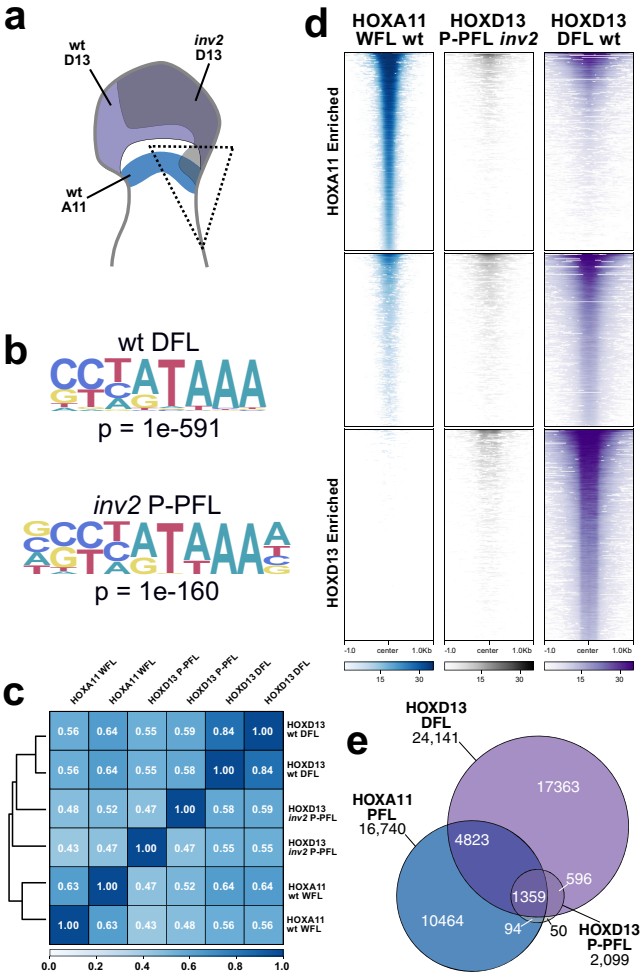

**Fig. 4 Ectopic HOXD13 binds only to positions bound by HOXD13 in the distal limb. a** Scheme as in Fig. 3, illustrating the dissection for this dataset. **b** HOXD13 consensus-binding motif found in E12.5 wild type distal forelimbs (DFL) and *inv2* posterior proximal forelimbs (P-PFL) demonstrates that the protein binds to the same sequences in the ectopic domain as it does in the normal distal limb domain, although in the *inv2* samples the motif was extended by one nucleotide on both sides. **c** Pearson correlation clustering on coverage from all replicates used for binding analysis demonstrating that the *inv2* P-PFL HOXD13 coverage most closely clusters with the wild type DFL HOXD13 dataset rather than the coverage of HOXA11 in proximal limbs. **d** Heatmaps of genomic regions differentially bound by HOXA11 (ref. [51]) at E11.5 in whole-forelimb samples (WFL) and HOXD13 at E12.5 in the distal forelimb of wild type embryos. Many of sites preferentially bound by HOXA11 are also bound by HOXD13 in the distal limb, but HOXD13 can also bind to many sites that HOXA11 does not bind to, as previously reported. The HOXD13-binding regions identified in the P-PFL tissue of *inv2* limbs were mapped onto these differentially bound regions. HOXD13 in the P-PFL binds to a portion of sites bound in the wild type distal limb. **e** Euler diagram of binding sites identified as significant peaks in both replicates of each experiment. The majority of HOX13-binding sites in the P-PFL overlap with HOXA11 sites. All samples are homozygous for the indicated genotype.

Together, these results support the conclusion that *Hoxd13* reduces the transcription of *Hoxa11* when expressed ectopically in proximal limb cells, yet to an amount that cannot account for the mesomelic dysplasia phenotype. The absence of a similar effect to *Hoxd11* in the *inv2* was not unexpected since *Hoxd13* and *Hoxd11* transcripts are often present in the same distal cells[47].

**Ectopic HOXD13-binding pattern in proximal limb cells**. In proximal limb cells, inappropriate expression of HOXD13 alters the normal expression of *Hoxa11* so it could also influence the expression of other genes important to normal limb formation. It may do this by binding to the same set of DNA sequences that the protein normally binds in distal cells[51], thus implementing a "distal" program in these proximal cells. Alternatively, HOXA11 and HOXD13 share very similar binding motifs and binding positions, yet they are normally expressed in mutually exclusive portions of the limb[46,51]. The co-expression of both factors in the same cells may redirect ectopic HOXD13 binding towards positions already bound by HOXA11. To discriminate between these possibilities, we analyzed HOXD13 binding in *inv2* mutant proximal limb cells by CUT&RUN[52]. The posterior proximal forelimb (P-PFL) region containing the *Hoxd13* ectopic domain was micro-dissected in duplicate (Fig. 4a, dashed triangle). The remaining portion of the limbs were processed for *Hoxd13* in situ hybridization as controls for the dissection (Supplementary Fig. 5a). To determine if HOXD13 binding is altered in the P-PFL, we also generated HOXD13 CUT&RUN from wild type distal forelimb cells and compared these with a previously reported HOXA11 whole-forelimb dataset, even though *Hoxa11* is expressed only in the proximal limb[51].

We first determined whether the HOXD13 consensus-binding motif was the same between the control DFL and the *inv2* P-PFL samples. De novo motif finding identified the previously reported motif[53–55] as the top-scoring motif in both control and *inv2* samples. In both samples, the HOXD13 motif was the only HOX motif identified among the top five high-scoring results, indicating that proximal forelimb chromatin environment did not alter its preferred binding sequence (Fig. 4b). We then performed a hierarchical clustering analysis of HOXD13-binding profiles found in the P-PFL sample to determine if they better match the HOXA11 or the HOXD13 profiles. We found that these samples cluster more closely with the wild type DFL-binding profiles than with the HOXA11 WFL samples (Fig. 4c). This was confirmed by differential binding analyses[56], where we determined all of the peaks bound preferentially by HOXD13 in control distal cells or by HOXA11 in proximal cells, or not bound preferentially at all, and compared with this the *inv2* P-PFL HOXD13 peaks. We observed a pattern of HOXD13 in P-PFL samples that most closely matched the control DFL HOXD13-binding profile although with a much lower signal (Fig. 4d and Supplementary Fig. 5b). Therefore, it appears that the set of HOXD13-binding sites identified in proximal cells expressing *Hoxd13* was closely related to the set of binding sites normally occupied by HOXD13 in distal cells, suggesting that the proximal limb cells may have undergone a partial transition to a distal limb identity.

Next, we looked at the percentage of those sites bound by HOXD13 either in control distal or in P-PFL cells, which would also be occupied by HOXA11 in proximal cells[51]. We compared the 24,141 HOXD13 peaks identified in the wild type DFL to the 16,740 HOXA11-binding sites identified in control proximal limbs and found that 6182 peaks (25% of HOXD13, 37% of HOXA11 peaks) overlapped between the two (Fig. 4e). We then evaluated the overlap between the HOXD13-binding sites in proximal cells and the distal forelimb. We found 1955 sites in common, but the proportion of those that overlap with HOXA11 had increased from 25% to 70% (1359). This change in proportion suggests that most of the sites uniquely bound by HOXD13 in the distal limb cannot be bound in the proximal limb, either because of the absence of essential co-factors, or because the small pool of HOXD13 factors was preoccupied at sites where HOXA11 is normally bound. Even though the scarcity of starting material may have introduced a technical bias, as

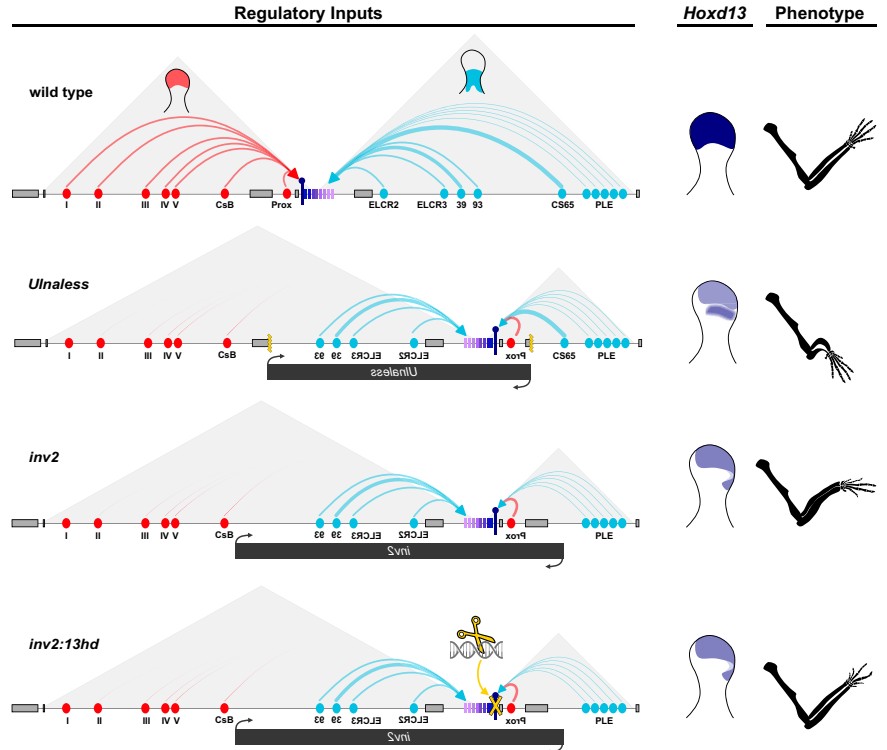

**Fig. 5 Schematic illustrations of the regulations at work in the various mouse alleles and their effect on *Hoxd13* expression.** On top is the control landscape, with distal limb enhancers (red) activating *Hoxd13* (purple pin) in distal cells. On the other side of the cluster, proximal limb enhancers (blue) regulate more telomeric *Hoxd* genes in proximal limb cells. As a result, *Hoxd13* is expressed in distal cells only (right), leading to the wild type phenotype. In the *Ulnaless* inversion, *Hoxd13* is positioned close to both a strong proximal enhancer (CS65) and a series of weaker enhancers (PLE). As a result, it becomes strongly expressed in proximal limb cells, while reduced in distal cells since it is separated from all distal limb enhancers except one (Prox), leading to a light phenotype in digits and strong mesomelic dysplasia. In the *inv2* inversion, *Hoxd13* is not adjacent to CS65 as in *Ulnaless*, but is now under the control of the weaker enhancer series (PLE). The gain of expression in proximal limb cells is weaker and accordingly, the mesomelic dysplasia is not as severe as in *Ulnaless* mice, while the digit phenotype is expectedly comparable. In *inv2:13hd* limb cells (bottom), the transcription of *Hoxd13hd* is the same as in *inv2*, yet the mesomelic dysplasia phenotype is completely rescued and the forearms are like wild type. However, because *Hoxd13* is now fully inactivated in distal cells, the digit phenotype is stronger and equivalent to a full *Hoxd13* knock-out[30].

illustrated by the total amount of detected peaks, this shift in proportions indicates a change in the binding repertoire towards a condition intermediate to interfering with HOXA11-binding sites and deployment of a distal limb program in the proximal limb cells. This is corroborated by the scRNA-seq experiment where the cells of proximal limb did not show a complete transition to distal limb cell identity.

## Discussion

***Hoxd13* as the cause of mesomelic dysplasia in mice**. In this study, we overcame the difficulty to use the mouse *Ulnaless* allele as a model system to understand the molecular etiology of mesomelic dysplasia by analyzing a similar yet less drastic condition where slightly different inversion breakpoints generated an ectopic gain of expression of *Hoxd13* in proximal cells, which was weaker and spatially more restricted than in the *Ulnaless* inversion. This weaker expression was due to the removal of most proximal limb enhancers (Fig. 5) leaving in place only a few weak proximal limb regulatory sequences. This group of proximal cells expressing *Hoxd13* was nevertheless large enough to induce a fully penetrant MD phenotype in homozygous mice, which otherwise could breed and were thus available for analysis. We took advantage of this to generate a secondary mutation in *cis* with the inversion, whereby a full loss-of-function of *Hoxd13* was induced. In these mice, the shortening and bending of bones associated with the primary mutation was fully rescued,

unequivocally demonstrating the central role of the gained *Hoxd13* in disrupting limb development.

Because the loss of function of both *Hoxa11* and *Hoxd11* induced severe mesomelic dysplasia in mice, it was argued using both murine and human conditions that various breakpoints around the *HOXD* cluster would lead to the down regulation of the latter two genes thus inducing bone anomalies[4,14]. Alternatively, it was proposed that while these two genes would remain transcribed, the ectopic presence of HOXD13 protein would functionally interfere with HOXD11 and HOXA11 proteins through a dominant-negative effect referred to as "posterior prevalence"[15,20]. While our datasets produced from both single-cell RNA sequencing and DNA-binding analyses of various HOX proteins do not allow us to completely discriminate between these two possibilities, it is clear that the former explanation alone cannot account for the observed MD phenotypes. Indeed, studies involving combined *Hoxa11/Hoxd11* mutations in mice observed mesomelic dysplasia phenotypes when at least half the total dose was removed[7]. In our single-cell RNA-seq experiment we observe in our *inv2* mice a 40–55% reduction of *Hoxa11* and a modest 15–30% reduction of *Hoxd11*, which together are not sufficient to elicit the described phenotype. In addition, the weak reduction in *Hoxd11* transcripts was not scored in our WISH where, if anything, a slight gain was observed.

Therefore, the partial reduction of *Hoxa11* transcription must be potentiated by another effect of ectopic HOXD13. Our differential binding analysis revealed that ectopic HOXD13 was

bound to a set of sites that most closely matched those bound by HOXD13 in control distal cells, with more than 25% of the binding sites identified in the proximal limb being sites normally restricted to the distal limb. This suggests that proximal cells may have acquired a more distal identity leading to a reduction in bone length. The recent observation that HOX13 proteins have a pioneer effect[51,57,58] provides a potential mechanism for this to occur. On the other hand, in proximal cells ectopically expressing HOXD13, 70% of those sites are normally bound by HOXA11, which suggests that a large part of the change in proximal cells may result from interactions between HOXD13 and HOX11 proteins at these binding sites, potentially through the dominant-negative effect of posterior prevalence. This may also apply to other circumstances where the gain of HOXD13 protein led to alterations identical to *Hox11* genes loss of function, for example, during the development of metanephric kidneys[7,59,60].

**An inclusive model for human mesomelic dysplasias at 2q31.** Thus far, none of the human mesomelic dysplasias mapping to 2q31 could be directly associated with mutations in *HOXD* genes, but breakpoints were identified and located in the vicinity of the *HOXD* cluster. As a consequence, the various reported deletions, inversions, or duplications were generally interpreted as inducing a regulatory loss-of-function to *HOXD* genes, in particular to *HOXD11* (refs. [1,4]). Additionally, the regulation of *Hoxd* genes during limb development has been shown to be evolutionarily conserved across amniotes[40] and it appears that many of the principles of genome architecture are similarly conserved between humans and mice[61,62]. We can thus revise these previous explanations and propose an inclusive model to account for various mesomelic dysplasias mapped at human 2q31, solely based on the abnormal gain of *HOXD13* transcription in proximal limb cells (Supplementary Fig. 6).

For instance, the duplication of the *HOXD* cluster reported in ref. [1] includes the ELCR2 proximal limb enhancer[63]. In the mutant chromosome, a copy of *HOXD13* is in close proximity with this enhancer, regardless of the orientation of the duplicated DNA sequence and will thus receive proximal regulatory inputs (Supplementary Fig. 6b, blue arrows), a fortiori since one of the duplicated copies of *HOXD13* is now located far from its digits-specific enhancers (red arrows), a situation that was shown in mice to de-sequester *Hoxd13* and re-allocate it towards proximal enhancers on the other side of the cluster[24]. In the case where the duplicated segment would also be inverted, *HOXD13* would be in even closer contact with multiple proximal limb enhancers (Supplementary Fig. 6b, top). A similar reasoning applies for the two families described in ref. [2]. In family 1, the duplicated and inverted copy of *HOXD13* is now in close contact with proximal enhancers (Supplementary Fig. 6c, top). Likewise, the large inverted duplication mapped in family 2 brings *HOXD13* even closer to proximal limb enhancers (Supplementary Fig. 6b, bottom).

Cho et al. (2010)[3] reported a family carrying a duplication extending across the *HOXD* cluster and the proximal limb regulatory landscape, without determining the orientation of the duplicated DNA segment (Supplementary Fig. 6d). In both orientations, however, the result is that the duplicated copy of *HOXD13* is de-sequestered from the distal regulatory landscape (red arrows) and is now licensed to interact with proximal enhancers (blue arrows). The same mechanism likely underlies the case reported by Peron et al. (2018)[4], which involves two deletions either in *cis* or in *trans*. If in *trans*, the deletion of all distal limb enhancers (Supplementary Fig. 6e, deletion 1) would again allow *HOXD13* to interact with proximal limb enhancers (Supplementary Fig. 6e, top). An almost identical configuration to

this had been previously shown in mice where the distal limb enhancers were removed through a large inversion (Supplementary Fig. 6e, bottom)[24]. If both deletions had occurred in *cis*, it is possible that the *HOXD* cluster, located in between, was inverted as well (Supplementary Fig. 6e, middle) (see ref. [28]). In this case, *HOXD13* would be positioned in the vicinity of proximal limb enhancers, leading to a strong gain of expression.

This explanatory framework can thus be applied to all reported cases of human mesomelic dysplasia associated with 2q31, so far without any exception. The various positions of *HOXD13* with respect to the proximal limb enhancers as observed in the different causative chromosomal rearrangements likely explains the variability in the strength of the alterations scored in the human forearms.

## Methods

**Animal work**. All experiments were approved and performed in compliance with the Swiss Law on Animal Protection (LPA) under license numbers GE 81/14 and VD2306.2 (to D.D.). All animals were kept in a continuous back cross with C57BL6 × CBA F1 hybrids. Sex of the embryos was not considered in this study. Mice were housed at the University of Geneva Sciences III animalerie with light cycle between 07:00 and 19:00 in the summer and 06:00 and 18:00 in winter, with ambient temperatures maintained between 22 and 23 °C and 45 and 55% humidity, the air was renewed 17 times per hour.

**Generation of the *HoxD^inv2* allele**. The *HoxD^inv2* allele was generated by STRING[29] using a cross between mice carrying the del(65) loxP allele[11] (the CS65 element was replaced with a loxP site), and mice carrying a loxP site inserted at chr2:74477755 (mm10) by a Sleeping Beauty transposon system[64]. F0 mice carrying these two loxP sites in *cis* were then crossed with the *Hprt*-Cre[65] and F1 animals from this cross were screened for the presence of the inversion between the coordinates chr2: 74477755–75438813 (mm10) with genotyping primers included in Supplementary Table 1. The regions of the inversion breakpoints were confirmed by Sanger sequencing. The founder mice were systematically backcrossed with CBA/C57B6 stocks for several generations, as for all HoxD alleles produced in the laboratory.

**Generation of the secondary *Hoxd13^hd* mutation**. We used a single CRISPR guide sequence that was used previously for the mutation of *Hoxd13* (Supplementary Table 1)[59]. The guide sequence was transcribed in vitro with NEB HiScribe T7 (NEB E2040S). The guide and TrueCut Cas9 v2 protein (Thermo Fisher A36497) were electroporated with a NEPA21 (NEPA GENE Co. Ltd, Chiba, Japan) into fertilized mouse embryos carrying the *HoxD^inv2* allele as previously reported[66]. Founders were screened by PCR for the presence of the inv2 allele and mutation to the *Hoxd13* second exon. Each founder was crossed with an allele containing a deletion of the *Hoxd* gene cluster to determine which founders contain the inv2 and *Hoxd13^hd* mutations in *cis* and we also screened for alleles containing the *Hoxd13^hd* mutation on the wild type chromosome to use as controls. A founder stock was identified for both alleles containing the same DNA mutation. Micro-CT scans were performed on littermates at 5 weeks after birth at the University of Geneva CMU and analyzed with Horos 3.3.6. Box plots for length measurements and *t*-test were produced in DataGraph 4.6.1.

**Histology, in situ hybridizations, and *lacZ* stains**. Embryos were collected at the indicated stages and processed following standard WISH procedures[67]. Embryos were treated with Proteinase K (EuroBio GEXPRK01-15): E10.5 and E11.5 at 1:2000 for 7 min, E12.5 at 1:1000 for 10 min, E13.5 at 1:1000 for 14 min, and E14.5 at 1:1000 for 40 min. For P3 skeletons, animals were collected at post-natal day 3. Alcian Blue and Alizarin Red stains were performed as previously reported[68]. *lacZ* stains were performed as previously reported[40]. Images of embryos were collected with an Olympus DP74 camera mounted on an Olympus MVX10 microscope using the Olympus cellSens Standard 2.1 software.

**ATAC-Seq**. Embryos were collected at E12.5 and placed in PBS on ice. Yolk sacs were collected, digested in buffer (10 mM EDTA pH 8.0 and 0.1 mM NaOH) at 95° for 10 min with shaking at 900 r.p.m. DNA from these samples was screened by genotyping with Z-Taq (Takara R006B). Two embryos were identified as homozygous for wild type or *HoxD^inv2* alleles and then were processed individually for ATAC-Seq. The proximal forelimbs of embryos were dissected and placed into PBS with 10% FCS and 8 μl collagenase at 50 mg/ml (Sigma C9697) at 37° for approximately 5 min. A single replicate of pooled cells for each tissue and each genotype were counted and 50,000 cells were isolated for processing with the Nextera Tn5 enzyme (Illumina FC-131-1096) as previously described[39]. Tn5-treated DNA was amplified with Nextera Library primers using NEBNext library amplification master mix (NEB M0541) and sequenced on an Illumina NextSeq.

Sequenced DNA fragments were processed as previously reported[58] with a minor modification on a local Galaxy server[69]: the BAM file was converted to BED prior to peak calling with bedtools version 2.18.2 (ref. [70]). Peak calling was done using MACS2 (v2.1.1.20160309) callpeak (--no-model --shift -100 --extsize 200 --call-summits --keep-dup all). For hierarchical clustering and heatmap analysis, two sets of bed files were generated. First, the peak regions were collected from the three wild type ATAC datasets, and then merged with bedtools (version 2.27.1) to remove redundant elements. The merged peaks in the region chr2: 73950000–75655000 excluding the *Lnpk* and *Mtx2* gene bodies as well as the region from *Evx2* to *Hoxd1* constitute the first set composed of non-genic elements. The second set contains the promoters (−1kb, +100 bp from TSS) that overlapped with at least one peak in a wild type ATAC dataset in the region from *Evx2* to *Hoxd1*. Heatmaps were generated with plotHeatmap from deepTools version 3.5 (ref. [71]). Clustering was performed with R (www.r-project.org) on matrices generated by multiBigWigSummary (deepTools version 3.5).

**CUT&RUN.** Posterior proximal forelimb cells were isolated and genotyped in the same manner as samples for the ATAC-Seq (above). Pools of cells from individual embryos (see Fig. 4a) were processed according to the CUT&RUN protocol[52] using a final concentration of 0.02% digitonin (Apollo APOBID3301). Cells were incubated diluted with 0.5 µg/100 µl of anti-HOXD13 antibody (Abcam ab19866), or 0.5 µg/100 µl of anti-CTCF (Active Motif 61311) at 4 °C. The pA-MNase was kindly provided by the Henikoff lab (Batch #6) and added at 0.5 µl/100 µl Digitonin Wash Buffer. Cells were digested in low calcium buffer and released for 30 min at 37 °C. Sequencing libraries were prepared with KAPA HyperPrep reagents (07962347001) with 2.5 µl of adaptors at 0.3 µM and ligated for 1 h at 20 °C. The DNA was amplified for 14 cycles. Post-amplified DNA was cleaned and size selected using 1:1 ratio of DNA:Ampure SPRI beads (A63881) followed by an additional 1:1 wash and size selection with HXB. HXB is equal parts 40% PEG8000 (Fisher FIBBP233) and 5 M NaCl. Sequenced DNA fragments were processed as previously reported[58] with slight modifications: PCR duplicates were removed with Picard before the BAM to BED conversion and in MACS2 using the option --keep-dup all instead of --keep-dup 1. Motif enrichment was performed on individual samples with HOMER version 4.10 (ref. [55]) using default conditions on peaks identified as significant from MACS2 in the second replicate. All samples were mapped to wild type mm10. The E11.5 whole-forelimb HOXA11 ChIP-Seq datasets (SRR8290670 of GSM3504924 and SRR8290672 of GSM3504925)[51] were down-sampled to 25mio reads with seqtk version 1.3 (https://github.com/lh3/seqtk/) using a RNG seed of 4 and then processed following a previously reported workflow[72]. Differential binding analysis was performed with DiffBind 2.14.0 (ref. [56]) on replicate sample peak sets identified by MACS2 for HOXD13 in wild type distal forelimb and HOXA11 in wild type forelimb, with default conditions using DESeq2 1.24.0 and R version 3.6. Hierarchical clustering analysis was performed with deepTools plotCorrelation, and the heatmap was generated with deepTools plotHeatmap.

**Capture Hi-C.** Samples used in the Capture Hi-C were identified by PCR screening embryos at E12.5 as described above. Collagenase-treated samples were cross-linked with 1% formaldehyde (Thermo Fisher 28908) for 10 min at room temperature and stored at −80° until further processing as previously described[40]. The SureSelectXT RNA probe design used for capturing DNA was done using the SureDesign online tool by Agilent. Probes cover the region chr2: 72240000–76840000 (mm9) producing 2× coverage, with moderately stringent masking and balanced boosting. Sequenced DNA fragments were processed as previously reported[40] but the mapping was performed on mm10 and the reads in chr2: 72402000–7700000 were selected. The mutant *inv2* genome was characterized from Sanger sequencing data around the inversion breakpoints. A custom R (www.r-project.org) script based on the SeqinR package[73] allowed the construction of a FASTA file for the inverted chromosome 2 from the wild type sequence and the exact position and sequence of breakpoints. The sequence of the mutant chromosome 2 (available at 10.5281/zenodo.4456654) was then compiled with other wild type chromosomes to form the mutant *inv2* genome. For samples that were mapped to the *inv2* mutant genome, the same workflow as described above was used. Heatmaps in Fig. 2 were plotted with pyGenomeTracks 3.3 (refs. [74,75]) and subtraction heatmaps in Figs. 2 and 2S were plotted with a custom tool available at https://github.com/lldelisle/scriptsForBoltEtAl2021. The TAD separation scores in Fig. 2S were computed with HiCExplorer hicFindTADs version 3.5.1.

**Enhancer transgenesis assay.** The enhancer regions used in the transgenesis assay (Fig. 2b and Supplementary Fig. 2e) were identified based on the presence of H3K27Ac histone modification and ATAC peaks in wild type proximal forelimbs, and also on the absence of CTCF. DNA sequences from these regions (Supplementary Table 1) were collected and assembled in silico to produce the PLE TgN sequence with *Kpn*I and *Apa*I restriction sites flanking the enhancer sequences. This 3.5 kb DNA sequence was synthesized by TWIST Bioscience (San Francisco, CA). The PLE sequence was restriction digested and ligated into the pSK-*lacZ* reporter construct. The 7.1 kb fragment containing the PLE:*lacZ* construction was excised from the vector backbone with the *Kpn*I and *Sac*II restriction enzymes, purified by agarose gel and column purification (Qiagen 28704). Pro-nuclear

injections were performed by the University of Geneva CMU. Embryos were collected at approximately E12.5 and stained for *lacZ*.

**Single-cell RNA-seq.** Embryos were collected and stored in 1× PBS treated with DEPC and held on ice while genotyping was performed (detailed above). Embryos with the desired genotype were selected and the posterior portion of each of the forelimbs was isolated for each replicate. The cells were digested in collagenase and stored in 1× PBS containing 10% FCS and 0.2 mM EDTA to prevent cellular aggregation. The cell samples were transferred to the EPFL Gene Expression Core Facility for preparation into 10X GEMs and reverse transcription according to the 10X Chromium 3.1 protocol. Sequencing reads were processed with Cell Ranger 3.1.0 for demultiplexing, barcode processing, and 3′ gene counting using a modified gene annotation file (10.5281/zenodo.4456702). Clustering analysis was performed with Seurat 3.2.3 with R 3.6.3. All commands used are available (https://github.com/lldelisle/scriptsForBoltEtAl2021). In order to correct the distribution of expression from sampling noise, we used a new method called baredSC (version 1.0.0 --minNeff 200)[45]. This allows us to evaluate a confidence interval of the fold-change between two conditions. This has been used in Supplementary Fig. 4b where median and 68% confidence interval are given. The results were processed in R (www.r-project.org) and plotted with ggplot2. In order to investigate the origin of clusters 7, 8, and 11, the scRNA-seq results from He et al.[43] were downloaded from https://cells.ucsc.edu/mouse-limb/10x/200120_10x.h5ad and imported as Seurat objects. Only cells corresponding to stages E12 and E13 were kept. Then, this subsetted dataset was integrated with our dataset using Seurat version 4.0.1 and with R version 4.0.

**Statistics and reproducibility.** All in situ hybridizations for all genes and for all genotypes were performed on at least three biological replicates each. In the PLE:*lacZ* transgenic assay we collected 34 viable embryos and 7 of them stained for lacZ; all of these embryos were included in Supplementary Fig. 2e. For skeleton preparations, three biological replicates were used for wild type and *inv2* mutants, measuring both forearms of each animal. For the *13hd* and *inv2:13hd*, two biological replicates were used, measuring both forearms of each animal. For CHiC-seq and CTCF CUT&RUN-seq, the tissue was pooled from six or seven embryos (both pairs of limbs) of the same genotype in order to produce enough material for experimentation. In the scRNA-seq, one wild type embryo was used, with the P-PFL collected from both forearms, and for each *inv2* sample, the same dissection was performed, in two distinct embryos. The HOXD13 CUT&RUN experiment was produced from two separate pools of cells dissected from multiple embryos (see Supplementary Fig. 5a).

**Reporting summary.** Further information on research design is available in the Nature Research Reporting Summary linked to this article.

## Data availability
The data generated in this study are available as raw and processed datasets in the Gene Expression Omnibus (GEO) repository under accession number "GSE165495". The mouse E11.5 HOXA11 ChIP-seq dataset was obtained from the GEO under SRR8290670 of "GSM3504924 [https://www.ncbi.nlm.nih.gov/sra?term=SRX5105273]" and SRR8290672 of "GSM3504925 [https://www.ncbi.nlm.nih.gov/sra?term=SRX5105274]". All other relevant data supporting the key findings of this study are available within the article and its Supplementary Information files or from the corresponding author upon reasonable request. Source data are provided with this paper.

## Code availability
All scripts necessary to reproduce figures from raw data (including custom scripts) are available at GitHub [https://doi.org/10.5281/zenodo.5118344][76].

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

## Acknowledgements

We thank Dr. Leonardo Beccari for early insights on this project, Dr. Josef Zakany for assistance with skeletal preparations and analysis, Dr. Alexandre Mayran for advice on single-cell RNA-seq and critical reading, as well as members of the Duboule laboratories for comments and discussion. We are grateful to Sandra Gitto and Thi Hanh Nguyen Huynh for their help with mice breeding. This work was supported in part using the resources and services of the Gene Expression Core Facility at the School of Life Sciences of EPFL. Part of the work was performed using the facilities of the Scientific IT and Application Support Center of EPFL. C.C.B was supported by the Eunice Kennedy Shriver National Institute of Child Health & Human Development of the National Institutes of Health, under Award Number F32HD093555. This work was supported by funds from the École Polytechnique Fédérale (EPFL, Lausanne), the University of Geneva, the Swiss National Research Fund (No. 310030B_138662), and the European Research Council grants SystemHox (No 232790) and RegulHox (No 588029) (to D.D.). Funding bodies had no role in the design of the study and collection, analysis and interpretation of data and in writing the manuscript.

## Author contributions

C.C.B.: designed and conducted experiments, analyzed datasets, formalized results, and wrote the paper. B.M.: designed, produced, genotyped, and helped analyze mouse mutants. L.D.: analyzed and evaluated the statistical significance of datasets, and wrote the paper. D.D.: designed experiments, transported mice, dissected some limb buds, and wrote the paper.

## Competing interests

The authors declare no competing interests.
