## [Peer Review File · Nature Communications]

Reviewers' Comments:

Reviewer #1:

Remarks to the Author:

The authors provide some detailed and elegant data on how an inversion of the mouse HoxD cluster likely mimics human Mesomelic Dysplasia. The combination of the inv2 mutant and a rescue of the dysplasia phenotype with a CRISPR-induced HOXD13 mutant is particularly elegant, and an important test of the hypothesis that the dysplasia phenotype is indeed due to the mis-expression of HOXD13. Also, the detail provided by combining Hi-C data with ATACseq data, and tying this all together with WISH expression data and phenotypes is very impressive. The detailed analysis of the single-cell sequencing data also gave impressive detailed insight alongside the binding site data, and is discussed in a balanced fashion with regards to the possibility for 'posterior prevalence'. In addition to providing a highly plausible explanation for the developmental genetic basis for human Mesomelic Dysplasia, this manuscript also provides important insights into vertebrate limb development more generally, as well as Hox gene regulatory mechanisms.

The entire manuscript is very well-written. The only things that I spotted were two extremely minor things.

First, near the bottom of page 7 the insulator boundary between Hoxd11 and 12 is mentioned. It seems odd to be talking about this boundary between 11-12 and then a couple of sentences later say that Hoxd12 is regulated in concert with Hoxd9-11, with only Hoxd13 being distinct.

Secondly, on line 487 change "would even be in closer" to, "would be in even closer".

Reviewer #2:

Remarks to the Author:

MESOMELIC DYSPLASIAS ASSOCIATED WITH THE HOXD LOCUS ARE CAUSED BY REGULATORY REALLOCATIONS

The manuscript describes a series of extensive experiments to test the hypothesis that misregulation of Hoxd13 is responsible for mesomelic dysplasias associated with the HoxD locus. The manuscript describes novel approaches to produce a unifying model for the aetiology of this defect.

The model of ectopic of HoxD13 in more proximal regions is an attractive explanation for the generation of mesomelic dysplasia. The inversion generated in the mouse mutant (inv2) and the resulting phenotype produced in the radius/ulna recapitulate characteristic features of the defect. Furthermore, the targeted deletion of HoxD13 in the background of the Inv2 model provides some strong supportive evidence for HoxD13 to be directly implicated in the phenotype generated.

The later analysis notwithstanding, the WISH analysis of the inv2 is difficult to reconcile with the proposed model. Hoxd13 expression levels appear markedly reduced in the Inv2 mutant. Furthermore, the apparent 'proximal' ectopic domain of HoxD13 is small, low level, and not very proximal. The domain indicated by the black arrow in 1d appears associated with the carpal region rather than around the cells that will form the radius and ulna. It is hard to imagine how the highlighted domain of Hoxd13 expressing cells could be responsible for the defects described in these bones. The Crispr-targeted deletion of Hoxd13 clearly resolves the mesomelic dysplasia, however

Line 89 normal forelimbs? Digits abnormalities are apparent in the compound Inv2/Hoxd13 del and the limb looks smaller overall. Are these compound mutant limbs really normal?

Given the ulnar deviation it might be expected that the ulna shortening is greater than that of the radius. Was this the case?

The concatenation of the proximal limb enhancers PLE01-05 is able to generate expression in a broader region of the zeugopod consistent with the observed mesomelic dysplasia. It remains puzzling why this is not supported by the WISH results

Line400- The binding patten is described to have a much lower signal. Is this to be expected since expression, as detected by WISH is apparently greatly reduced?

Line 402 "proximal cells may have undergone a partial transition to distal limb identity" Could this be assessed? As shown, HoxD13 is a 'classic' distal marker but expression is actually reduced in the Inv2 mutant. IS this really borne out by the dysplasia phenotype?

Minor

LnPk annotation is hard to see on Figure 1

Reviewer #3:

Remarks to the Author:

The manuscript by Bolt et al. reports the molecular consequences of structural variations associated with mesomelic dysplasia at the HOXD locus. The authors create a new mouse model (inv2), harbouring a large inversion at the HoxD locus, which is reminiscent of the ulnaless mutant mouse. Mice homozygous for the inv2 allele are viable and develop a mesomelic dysplasia phenotype in the forelimbs. By investigating chromatin interactions (Capture-HiC) the authors demonstrate that Hoxd13 ectopically interacts with proximal enhancers and they functionally validate them in transgenic mice. Ectopic contacts in the inv2 allele lead to misexpression of Hoxd13 in the posterior proximal forelimb. Importantly, functional loss of Hoxd13 on the inv2 allele rescues the mesomelic dysplasia phenotype, demonstrating HOXD13 as driver of the disease. Employing WISH and scRNA-seq in wt and inv2 limbs, the authors disentangle the molecular consequences of Hoxd13 expression in the posterior proximal forelimb on expression levels of other Hox genes. The analysis reveals that a previously suggested combined downregulation of Hoxa11 and Hoxd11 cannot account for the disease phenotype. Instead, and in conjunction with ChIP-seq for HOXD13 in posterior proximal forelimbs of inv2 animals, the authors suggest the adoption of a HOXD13-mediated transcriptional program in proximal limb cells. Finally, based on the identified pathomechanism induced by inversions (inv2 and ulnaless alleles) the authors lay out a conceptual framework for the interpretation of multiple complex structural variations at the HOXD locus associated with the mesomelic dysplasia phenotype in patients.

The manuscript is well written and structured. The major conclusions and experiments are based on robust genetic evidence by the engineered inv2 and inv2:13hd mouse alleles. The data are well-presented and conclusions appear supported by the data. The inversion-induced reallocation of genes and enhancers and the thereafter identified gain-of-expression of Hoxd13 helps to interpret structural variations at the HOXD locus that have been identified in clinical settings. Importantly, the manuscript also addresses the molecular consequences of a misexpressed transcription factor to understand a disease etiology. This aspect is indeed not well covered by the current literature and makes the manuscript relevant to a broader readership. I do not think the manuscript needs further experiments, though I encourage the authors to address the following points to strengthen conclusions of the manuscript.

Comments (sorted by relevance):

1. The cell clusters derived from scRNA-seq revealed cluster 7, 8 and 11 to be expanded in the inv2 mutant limb when compared to wt. The reader is left with the question where this obvious difference comes from? Following the argumentation in the manuscript, these clusters have a proximal limb identity. It is important for clarity and validity of the data to clarify in the manuscript whether this difference is of technical nature or can be attributed to changes in the inv2 mutants.

2. As pointed out by the authors themselves, the HOXD13 ChIP-seq in P-PFL appears very weak, questioning the quality of the data. From my point of view, the obtained signal at several loci in P-PFL is sufficient for the conclusion made in the manuscript. For better judgment by the reader, I

would suggest including a snapshot of a genomic locus showing ChIP-seq enrichment of HOXD13 in DFL and P-PFL and, if possible, differential binding of HOXA11.

3. The *ulnaless* and *inv2* breakpoints vary and the authors speculate that the inclusion of the proximal enhancer 65 in the T-DOMneo of the *ulnaless* allele could be responsible for stronger ectopic expression of *Hoxd13* and phenotype in *ulnaless* mutants (Summary figure 5). What happens to enhancer 65 in the *inv2* allele? Does the inversion breakpoint partially or completely disrupt the enhancer 65? The difference map in Figure 2b shows some enrichment in this region. A statement about what happens with this enhancer would clarify this, e.g. pointing to the *inv2* ATAC data, where the ATAC signal at enhancer 65 seems to be gone.

4. Are the ATAC data in replicates or singletons? I could not find a reference clarifying this in the manuscript or the reporting summary. The GEO entry suggests singletons. Please indicate in the manuscript. The ATAC data derived from the different limb areas look reproducible and I think that singleton experiments are sufficient for the type of analysis and interpretation of the data in the manuscript.

5. In Figure 4 the authors use previously published ChIP-seq data set for HOXA11 from Desanlis et al., 2020. Figure 4b, c, d, figure descriptions and corresponding main text switches between HOXA11 ChIP-seq from whole fore limb (WFL) and proximal forelimb (PFL). A more consistent naming will increase readability. Also refer to the used dataset using GEO entries GSM3504924 and GSM3504925. The given accession numbers (SRR8290670 and SRR8290672) in the method part did not allow me to immediately identify the used datasets to clarify this.

6. The regulatory consequences of structural variations, including inversions have been studied at multiple other loci and from different research groups. Furthermore, enhancer-gene reallocation and rearrangements relative to CTCF-sites and TAD structure have been extensively shown in other disease contexts such as cancer and other congenital disease with limb defects. To help their readers, I would strongly recommend the authors to include/cite additional relevant publications to better relate this work with the current literature.

7. Supplementary Figure 3d is impossible to read in print.

8. For better clarity I recommend using a unified nomenclature for indicating hetero- and homozygous state as in line 187 also for the figure description of Figure 1 and Supplementary Figure 1.

9. In line 248, introduce the abbreviation of PL here.

Authors' Response to Reviewers Comments for:

NCOMMS-21-05998-T

**Mesomelic dysplasias associated with the *HoxD* locus are**
**caused by regulatory reallocations**

Christopher Chase Bolt, Lucille Lopez-Delisle, Bénédicte Mascrez,
and Denis Duboule

We would like to thank the reviewers for the effort and detail that they have included in their
evaluation of our work, as well as for their globally positive comments. In the following pages
we have provided a point-by-point response to all comments and concerns, which are
emphasized in italics. We hope that these changes sufficiently address the referee's comments
and improve the quality of the manuscript.

In order to make the reading easier, we have included the referee's comments *in extenso* in
italics (small font), whereas our replies are non-italicized. We have also included at the end a
figure, which we do not think is essential to include in the paper, unless referees think
otherwise. We have also taken this opportunity to slightly improve the text.

Christopher Chase Bolt and Denis Duboule

**Reviewer #1 (Remarks to the Author):**

*The authors provide some detailed and elegant data on how an inversion of the mouse HoxD cluster likely mimics*
*human Mesomelic Dysplasia. The combination of the inv2 mutant and a rescue of the dysplasia phenotype with a*
*CRISPR-induced HOXD13 mutant is particularly elegant, and an important test of the hypothesis that the*
*dysplasia phenotype is indeed due to the mis-expression of HOXD13. Also, the detail provided by combining Hi-*
*C data with ATACseq data, and tying this all together with WISH expression data and phenotypes is very*
*impressive. The detailed analysis of the single-cell sequencing data also gave impressive detailed insight*
*alongside the binding site data, and is discussed in a balanced fashion with regards to the possibility for 'posterior*
*prevalence'. In addition to providing a highly plausible explanation for the developmental genetic basis for human*
*Mesomelic Dysplasia, this manuscript also provides important insights into vertebrate limb development more*
*generally, as well as Hox gene regulatory mechanisms.*

*The entire manuscript is very well-written. The only things that I spotted were two extremely minor things.*

*First, near the bottom of page 7 the insulator boundary between Hoxd11 and 12 is mentioned. It seems odd to be*
*talking about this boundary between 11-12 and then a couple of sentences later say that Hoxd12 is regulated in*
*concert with Hoxd9-11, with only Hoxd13 being distinct.*

We thank this referee for this positive comment and we do understand the confusion. The
insulation boundary observed within the *HoxD* cluster is dynamic, changing position during
the process of limb development as shown in Andrey et al., 2013 (Science) and Rodriguez-

Carballo et al. 2017 (Genes & Development). The factor most relevant to this study is that in
proximal limb cells, the boundary insulates *Hoxd13* from the active proximal limb enhancers.
It also insulates *Hoxd12* but in each well done WISH staining of the latter gene one can start
to see a ‘patch’ posteriorly, which indicates that the regulation can ‘touch’ this gene that likely
serves as a last defense against activating *Hoxd13* (interestingly, *Hoxd12* has disappeared from
snakes..).

To improve this, we have modified the text by removing the portion of the sentence, “separating
*Hoxd13* and *Hoxd12* from the other genes of the cluster”, as we realized it added unnecessary
complexity to the story without improving clarity (Line...)

*Secondly, on line 487 change “would even be in closer” to, “would be in even closer”.*

Thanks. We have made this recommended change to the text.

**Reviewer #2 (Remarks to the Author):**

*The manuscript describes a series of extensive experiments to test the hypothesis that misregulation of *Hoxd13* is*
*responsible for mesomelic dysplasias associated with the *HoxD* locus. The manuscript describes novel approaches*
*to produce a unifying model for the aetiology of this defect.*

*The model of ectopic of *HoxD13* in more proximal regions is an attractive explanation for the generation of*
*mesomelic dysplasia. The inversion generated in the mouse mutant (*inv2*) and the resulting phenotype produced*
*in the radius/ulna recapitulate characteristic features of the defect. Furthermore, the targeted deletion of *HoxD13**
*in the background of the *Inv2* model provides some strong supportive evidence for *HoxD13* to be directly*
*implicated in the phenotype generated.*

*The later analysis notwithstanding, the WISH analysis of the *inv2* is difficult to reconcile with the proposed model.*
**Hoxd13* expression levels appear markedly reduced in the *Inv2* mutant. Furthermore, the apparent ‘proximal’*
*ectopic domain of *HoxD13* is small, low level, and not very proximal. The domain indicated by the black arrow*
*in 1d appears associated with the carpal region rather than around the cells that will form the radius and ulna.*
*It is hard to imagine how the highlighted domain of *Hoxd13* expressing cells could be responsible for the defects*
*described in these bones.*

First, we would like thank the reviewer for the positive and detailed comments and serious
consideration of the underlying biology. There are two points made in this comment so we will
address each in series.

*“*Hoxd13* expression levels appear markedly reduced in the *Inv2* mutant.”*

This decrease is of course fully expected following the various reallocations of enhancers.
While the focus of this work was primarily on the role of *Hoxd13* in the proximal limb, the
inversion also affects the distal limb by disconnecting *Hoxd13* from the C-DOM (the TAD
containing distal enhancers). We state that *Hoxd13* is reduced in the distal limb domain and
produces a digit defect there (e.g., lines 176, 210, 267 among others) in the manuscript, but this
comment suggest that we may have been clearer in explaining this issue. It is critical to note

that there are two distinct but interrelated events happening in the limbs of *inv2* mutants. The
first is that the inversion moves *Hoxd13* (as well as *Hoxd12*, *d11*, and *d10*) away from their
normal distal limb digit enhancers causing a strong reduction in *Hoxd13* transcription in the
*distal* limb (only the Prox enhancer is inverted along, which accounts for the detected
expression level). The second is that the inversion also moves *Hoxd13* close to (a) proximal
limb enhancer(s), causing an ectopic transcription in a small spot at the distal and posterior
aspect of the *proximal* limb segment. It is thus expected that in the two domains of *Hoxd13*
expression, the transcript levels should be much below the control level.

The reviewer then expressed concern that these cells are actually within the normal distal or
carpal domain of the limb, not the proximal limb:

“the apparent ‘proximal’ ectopic domain of *HoxD13* is small, low level, and not very proximal.”

We of course agree with the observation that the domain is ‘small’ and ‘low level’ (see above)
and we argue in the manuscript that this “*small, low level ... expression*” is indeed the reason
for the weak mesomelic dysplasia compared with the strong MD observed in *Ulnaless* mice.
As explained in the introduction, this inversion was used *because* of this weak phenotype, since
*Ulnaless* mice could not be bred and hence the series of experiments presented here would have
been virtually impossible to carry out.

However, we disagree that these cells are digit or carpal cells as made clear by the WISH
experiments in Figure 3a. In this figure it is clear that the ectopic *Hoxd13* expression domain
in the proximal limb is at the same position as *Hoxa11* and *Hoxd11*, which are proximal limb
markers and which elicit a zeugopodial phenotype when inactivated. The expression domains
of these genes are highly relevant to our analysis exactly because of the relationship between
them and the ectopic expression of *Hoxd13* and the phenotype. To try to be more convincing
to this reviewer, we have reproduced this figure into an additional figure attached at the end of
this document, with annotations that draw attention to this detail as well as indicate that this
position is in the proximal limb domain at the same position as *Hoxd11* and *Hoxa11* genes (See
Figure for Reviewer). We understand that this point is not obvious to admit when first looking
at a WISH of the *Hoxd13* gain of function, but this patch of cells is identical to the posterior
patch of cells expressing for example *Hoxd4* (see figure). Since *Hoxd4* is expressed at low
levels in the proximal domain (same enhancers than group 10 and 11, though) one can clearly
see this patch located at the exact same position.

This observation was sufficiently clear to us that it became the foundation for performing the
follow-up experiments with the time-course WISH (Supplemental Figure 3a) and by an
orthogonal approach using single-cell RNA-seq experiment (Figure 3b-f). Both of these
experiments support and expand on the finding that the ectopic *Hoxd13* expression in the
proximal limb interferes partly with the expression and function of *Hoxa11/Hoxd11* in the
formation of the MD phenotype.

Finally, this referee states:

*The Crispr-targeted deletion of Hoxd13 clearly resolves the mesomelic dysplasia, however.*

Yes, this is of course the key data, which functionally demonstrates that this patch of cells (the
only place in the animal where *Hoxd13* is gained) causes the MD.

*Line 89 normal forelimbs? Digits abnormalities are apparent in the compound Inv2/Hoxd13 del and the limb*
*looks smaller overall. Are these compound mutant limbs really normal?*

We believe that the reviewer's comment refers to the text between 172 and 189 not "Line 89"
as there is no reference to normal or abnormal limbs at line 89 relevant to this context, so we
will direct our response to the text at line 189.

In fact, we do observe *digit* abnormalities in the compound *inv2/Hoxd13^{hd}* homozygous
mutants and we state this in the text, "While mice homozygous for this *Hoxd13^{hd}* mutation in-
*cis* with the *HoxD^{inv2/inv2}* inversion (*HoxD^{inv2/inv2}:Hoxd13^{hd/hd}*) also displayed the expected loss-
of-function phenotype in their digits..." In the manuscript we wrote that we expected a severe
digit phenotype because the individual *Hoxd13* loss-of-function phenotype produces digit
defects (Dolle *et al.*, 1993) and, in the *HoxD^{inv2/inv2}:Hoxd13^{hd/hd}* configuration, these limbs are
missing a functional copy of *Hoxd13* as well as a having a reduction in the expression of other
*Hoxd* genes (see Supplementary Figure 2c).

Regarding the second part of the question, we also quantified the lengths of the radius and ulna
*forelimb* bones by micro-CT for multiple animals of all the genotypes reported here, which can
be seen in Figure 1c. Variation in bone length does exist between individuals, such as the one
selected for representation in Figure 1b, but when we quantified the lengths of forelimb bones
in multiple limbs of each genotype and evaluated the variation by the Welch's t-test, we found
no significant difference in the mean length of the forelimb bones between wild type and
*inv2/13hd* rescue animals. We refer the reviewer to Supplementary Video 1 for a clear three-
dimensional representation of the changes observed and Figure 1c for quantitation of these
differences.

*Given the ulnar deviation it might be expected that the ulna shortening is greater than that of the radius. Was this*
*the case?*

Please see the response to the question above for details. We observed a reduction of the length
of the radius and ulna of approximately 20% in the *inv2* mutants. See Figure 1c for quantitation.

*The concatenation of the proximal limb enhancers PLE01-05 is able to generate expression in a broader region*
*of the zeugopod consistent with the observed mesomelic dysplasia. It remains puzzling why this is not supported*
*by the WISH results.*

In addition to the elements tested in this study, there are other putative enhancer elements that
were not tested, that could be acting as proximal limb enhancers in the *inv2* mutant limbs. As
we described in the manuscript, in some genetic configurations of the *HoxD* locus, we have
observed some enhancers that lie beyond the distal T-DOM insulation boundary that are
capable of driving *Hoxd* gene expression in the proximal limb portion where ectopic *Hoxd13*

is observed in *inv2* limbs by WISH (see Figure 7 in Yakushiji-Kaminatsui *et al.*, eLife, 2018).
Unfortunately, we cannot test all of these elements as individual transgenes, so we took an
approach that we believed offered the greatest likelihood of success, the results of which are
seen in Figure 2b and Supplementary Figure 2e. As for why these elements - individually or as
concatenates - produce a staining pattern that is slightly different than the WISH remains
unresolved, but may result from the fact that they are randomly integrated genomic fragments
and not within their normal coordinated regulatory landscape. The reviewer will find an
extended discussion on this phenomenon in (Bolt and Duboule, Development, 2020).
Regardless of these differences in the transgene assays, which lack experimental controls, the
rescue experiment presented in Figure 1 demonstrates that *Hoxd13* and its ectopic expression
are central to the phenotype.

*Line400- The binding pattern is described to have a much lower signal. Is this to be expected since expression, as*
*detected by WISH is apparently greatly reduced?*

This is a valid point and there are several potential explanations for the low signal observed for
HOXD13 in the P-PFL samples, which are enumerated here: (1) Globally, the *inv2* mutation
reduced *Hoxd13* transcription in the distal limb but weakly increased it in the proximal limb,
which is likely to influence the quantity of protein and so protein-DNA interactions are possibly
very few in these cells. (2) The number of cells used in the HOXD13 P-PFL samples are
significantly fewer than in the control DFL samples. (3) The two HOXD13 P-PFL samples
have increased background compared to the wild type DFL samples. This makes peak-calling
less accurate and contributes to weaker signal in the heatmaps in Figure 4d. (4) All of these
effects may contribute to the observed binding pattern in P-PFL. To accommodate this
weakness in the data and improve robustness, the experiment was performed on separate pools
of embryonic limbs, and performed in experimental replicates.

*Line 402 “proximal cells may have undergone a partial transition to distal limb identity” Could this be assessed?*
*As shown, HoxD13 is a ‘classic’ distal marker but expression is actually reduced in the Inv2 mutant. IS this really*
*borne out by the dysplasia phenotype?*

If we understand well, the reviewer asks whether it is possible to assess if proximal limb cells
acquire a distal identity? We think there is sufficient data in this manuscript to support our
inference that the *inv2* allele produces a small number of proximal limb cells which are partially
converted to a distal limb identity and are causal to the phenotype. The first evidence is from
the single-cell RNA-seq experiment in Figure 3 where we show that wild type cells with
proximal limb identity acquire *Hoxd13* transcription (cluster 6) in the *inv2* configuration. This
is most clearly demonstrated by the influence of ectopic *Hoxd13* in cluster 6 to repress the
transcription of *Hoxa11* (Figure 3a and 3f, and Supplementary Figure 3a). The second evidence
is from the HOXD13 CUT&RUN experiment. Here we observed a disproportionately large
number of HOXD13 binding sites that overlap with HOXA11 in the P-PFL compared with the
HOXD13 in the DFL (Figure 4e). Importantly, we do not conclude that the *inv2* allele causes
a conversion of the proximal limb to digit identity, but rather there is minor shift in the
transcriptional program of the P-PFL cells caused by the distal limb identity gene *Hoxd13*.

With regard to the relationship between ectopic *Hoxd13* expression and the MD phenotype, it
is important to distinguish the reduction of *Hoxd13* in cells with distal limb identity from the
increase of *Hoxd13* in cells with proximal limb identity. We observed a clear and reproducible
increase of *Hoxd13* in the proximal limb cluster 6. In conjunction with the rescue allele
experiment, we feel it is clear that the dysplasia phenotype arises directly from ectopic
expression of *Hoxd13* in the proximal limb.

*Minor*

*Lnpk annotation is hard to see on Figure 1*

Thanks. We have corrected this by making the annotation larger.

**Reviewer #3 (Remarks to the Author):**

*The manuscript by Bolt et al. reports the molecular consequences of structural variations associated with*
*mesomelic dysplasia at the HOXD locus. The authors create a new mouse model (inv2), harbouring a large*
*inversion at the HoxD locus, which is reminiscent of the ulnaless mutant mouse. Mice homozygous for the inv2*
*allele are viable and develop a mesomelic dysplasia phenotype in the forelimbs. By investigating chromatin*
*interactions (Capture-HiC) the authors demonstrate that Hoxd13 ectopically interacts with proximal enhancers*
*and they functionally validate them in transgenic mice. Ectopic contacts in the inv2 allele lead to misexpression*
*of Hoxd13 in the posterior proximal forelimb. Importantly, functional loss of Hoxd13 on the inv2 allele rescues*
*the mesomelic dysplasia phenotype, demonstrating HOXD13 as driver of the disease. Employing WISH and*
*scRNA-seq in wt and inv2 limbs, the authors disentangle the molecular consequences of Hoxd13 expression in the*
*posterior proximal forelimb on expression levels of other Hox genes. The analysis reveals that a previously*
*suggested combined downregulation of Hoxa11 and Hoxd11 cannot account for the disease phenotype. Instead,*
*and in conjunction with ChIP-seq for HOXD13 in posterior proximal forelimbs of inv2 animals, the authors*
*suggest the adoption of a HOXD13-mediated transcriptional program in proximal limb cells. Finally, based on*
*the identified pathomechanism induced by inversions (inv2 and ulnaless alleles) the authors lay out a conceptual*
*framework for the interpretation of multiple complex structural variations at the HOXD locus associated with the*
*mesomelic dysplasia phenotype in patients.*

*The manuscript is well written and structured. The major conclusions and experiments are based on robust genetic*
*evidence by the engineered inv2 and inv2:13hd mouse alleles. The data are well-presented and conclusions*
*appear supported by the data. The inversion-induced reallocation of genes and enhancers and the thereafter*
*identified gain-of-expression of Hoxd13 helps to interpret structural variations at the HOXD locus that have been*
*identified in clinical settings. Importantly, the manuscript also addresses the molecular consequences of a*
*misexpressed transcription factor to understand a disease etiology. This aspect is indeed not well covered by the*
*current literature and makes the manuscript relevant to a broader readership. I do not think the manuscript needs*
*further experiments, though I encourage the authors to address the following points to strengthen conclusions of*
*the manuscript.*

We would like to thank the reviewer for his/her careful and complete reading of the manuscript
as well as for helpful suggestions.

*Comments (sorted by relevance):*

*1. The cell clusters derived from scRNA-seq revealed cluster 7, 8 and 11 to be expanded in the inv2 mutant limb*
*when compared to wt. The reader is left with the question where this obvious difference comes from? Following*
*the argumentation in the manuscript, these clusters have a proximal limb identity. It is important for clarity and*

*validity of the data to clarify in the manuscript whether this difference is of technical nature or can be attributed*
*to changes in the inv2 mutants.*

This comment is well taken. It is an important point and one that we have been considering for
some time. We think that there is a technical explanation for why this may be. The ectopic
patch of *Hoxd13* expressing cells is small and very difficult to dissect in an identical manner
each time. Indeed, there is no real morphological landmark or even a tool that can be used to
guarantee reproducibility of this micro dissection. If by chance our dissection of the wild type
was slightly smaller than what was in the replicates of the *inv2* samples - excluding a portion
of the nascent proximal limb bone chondrogenic primordia - then this may appear as cell types
that are not present in the control sample. During the peer-review process we have evaluated
some recently published whole-limb single-cell RNA-seq experiments in parallel with our
datasets and found that the cells from clusters 7, 8, and 11 are present in the whole forelimbs
of wild type embryos, strongly indicating that the difference comes from our dissections and is
not a consequence of the inversion. We agree with the reviewer that this is an important element
in the analysis so we have generated a new panel for Supplementary Figure 3b demonstrating
the presence of these cells in the whole limbs of wild type embryos. We have also added text
to the manuscript beginning at line 320 to indicate that these clusters are normal and are likely
to appear different due to dissection variation.

*2. As pointed out by the authors themselves, the HOXD13 ChIP-seq in P-PFL appears very weak, questioning the*
*quality of the data. From my point of view, the obtained signal at several loci in P-PFL is sufficient for the*
*conclusion made in the manuscript. For better judgment by the reader, I would suggest including a snapshot of a*
*genomic locus showing ChIP-seq enrichment of HOXD13 in DFL and P-PFL and, if possible, differential binding*
*of HOXA11.*

Again, we agree with all of these points. A detailed explanation for the weak HOXD13
CUT&RUN signal in the P-PFL samples is provided above at line 190 in response to Reviewer
#2. To overcome this limitation in the data we produced biological replicates of the experiment
– each replicate containing a pool of dissected tissues - and used the peaks common to both
replicates in our analysis in order to improve experimental robustness. We agree that it will be
helpful to readers to see the differential binding in genomic tracks, so we have created a new
panel for Supplementary Figure 5 that contains screenshots of the different combinations of
binding sites between the three conditions.

*3. The ulnaless and inv2 breakpoints vary and the authors speculate that the inclusion of the proximal enhancer*
*65 in the T-DOMneo of the ulnaless allele could be responsible for stronger ectopic expression of Hoxd13 and*
*phenotype in ulnaless mutants (Summary figure 5). What happens to enhancer 65 in the inv2 allele? Does the*
*inversion breakpoint partially or completely disrupt the enhancer 65? The difference map in Figure 2b shows*
*some enrichment in this region. A statement about what happens with this enhancer would clarify this, e.g.*
*pointing to the inv2 ATAC data, where the ATAC signal at enhancer 65 seems to be gone.*

In the generation of the del65 allele used to generate the *inv2* allele (for details see Andrey *et*
*al.*, Science, 2013), the CS65 enhancer element has been replaced with a LoxP site which is
why there are no reads mapping to this position in the ATAC data. We do not put emphasis on
this detail because it is explained in the referenced publication, but more importantly because

CS65 does not remain in the T-DOMneo after inversion, which we presume would contribute
to strong *Hoxd13* in the proximal limb, such as seen in the *Ulnaless* allele. We have added a
statement to the Methods section that specifies that CS65 was deleted in the generation of the
del65 allele, line 817.

*4. Are the ATAC data in replicates or singletons? I could not find a reference clarifying this in the manuscript or*
*the reporting summary. The GEO entry suggests singletons. Please indicate in the manuscript. The ATAC data*
*derived from the different limb areas look reproducible and I think that singleton experiments are sufficient for*
*the type of analysis and interpretation of the data in the manuscript.*

The ATAC data are in singletons since, as the referee, we thought it was not necessary to
generate a duplicate for the use we make out of this dataset. We apologize for this oversight.
We have added this detail into the Methods section, line 856.

*5. In Figure 4 the authors use previously published ChIP-seq data set for HOXA11 from Desanlis et al., 2020.*
*Figure 4b, c, d, figure descriptions and corresponding main text switches between HOXA11 ChIP-seq from whole*
*fore limb (WFL) and proximal forelimb (PFL). A more consistent naming will increase readability. Also refer to*
*the used dataset using GEO entries GSM3504924 and GSM3504925. The given accession numbers (SRR8290670*
*and SRR8290672) in the method part did not allow me to immediately identify the used datasets to clarify this.*

We apologize for this mistake with the GEO entries and thank you for noticing. With regard to
the naming of the HOXA11 samples, we agree that consistent naming changes will improve
readability. There were some mistakes in the text and figures. We have added text to specify
that the dissection used for HOXA11 ChIP-Seq used whole forelimbs, but the expression
domain is in the proximal forelimb. Lines 379 and 389.

*6. The regulatory consequences of structural variations, including inversions have been studied at multiple other*
*loci and from different research groups. Furthermore, enhancer-gene reallocation and rearrangements relative*
*to CTCF-sites and TAD structure have been extensively shown in other disease contexts such as cancer and other*
*congenital disease with limb defects. To help their readers, I would strongly recommend the authors to include/cite*
*additional relevant publications to better relate this work with the current literature.*

We appreciate the reviewer's request that we should include references to recent works
evaluating the impact of 3D-genome structure on gene expression. However, we also tried to
remain focused on the mechanism at work in this particular inversion, and while we did perform
experiments that capture chromatin conformation, these were not central to the interpretation
of the results. The inversion of the CTCF sites at *HoxD* was a consequence of inversion, not
the test, and the interpretations made here would largely remain the same, should TAD and
CTCF be unknown. In fact, in contrast to many recent publications, this work did not aim at
reproducing a perturbed human genotype in mice or to test the importance of CTCF sites.
Instead, it is a part of an ongoing body of work to evaluate the mechanics of *HoxD* regulation
and function. We certainly agree that many of the studies suggested by the reviewer have been
very insightful and we have included references to some of these works because they
demonstrate that many of the principles of genome organization are conserved from humans to
mice which have allowed us to extrapolate how the human 2q31 genotypes are likely to form

the MD phenotype. Line 484. Should this expert think that a particular reference to a precise
publication is still missing, we would be happy to include it.

*7. Supplementary Figure 3d is impossible to read in print.*

Apologies, we share the reviewers concern here and hence we have split the content of
(previously) Supplementary Figure 3 into two; Supplementary Figure 3 and Supplementary
Figure 4. To accommodate this change we have moved the content of (previously)
Supplementary Figure 4 to a Supplementary Figure 5, and the content of (previously) S. Figure
5 to S. Figure 6.

*8. For better clarity I recommend using a unified nomenclature for indicating hetero- and homozygous state as*
*in line 187 also for the figure description of Figure 1 and Supplementary Figure 1.*

We apologize that this is not clear. All of the embryos used in this study, with the exception of
the PLE lacZ transgenic embryos, are homozygous for the indicated alleles. We have added
text to each relevant figure legend to make this explicit.

*9. In line 248, introduce the abbreviation of PL here.*

Thank you. It is corrected.

**Figure to address the concern of Reviewer 2.** (a) An enlargement of Figure 3a from the
manuscript with additional annotations. We have drawn lines across the three panels of the *inv2*
samples to demonstrate that the ectopic *Hoxd13* expression domain coincides with the normal
proximal limb segments expressing *Hoxd11* (and *Hoxd9* and *Hoxd4*) and *Hoxa11*. The solid
black line at the top shows that the limbs are aligned to the same distal tip of the limbs. (b)
Whole mount in situ hybridizations for *Hoxd11*, *Hoxd9*, and *Hoxd4* on wild type limbs showing
various intensities of the ‘proximal’ expression domain(s). The black arrow points to the exact
same region where *Hoxd13* is ectopically expressed in *inv2* mutant limbs, right below the
inflection point between the zeugopod and the autopod and at extreme posterior aspect of the
future zeugopod.

Reviewers' Comments:

Reviewer #1:

Remarks to the Author:

Thank you for accommodating my few minor suggestions. I have no further comments or concerns.

David Ferrier

Reviewer #2:

Remarks to the Author:

The authors have addressed the reviewers comments extensively

Reviewer #3:

Remarks to the Author:

Dear authors,

I would like to congratulate you on your manuscript. My questions and comments have been fully addressed and implemented in your newly revised manuscript. Therefore I am positive for publication.

**Reviewers' Comments**

*Reviewer #1 (Remarks to the Author):*

*Thank you for accommodating my few minor suggestions. I have no further comments or concerns.*

*David Ferrier*

*Reviewer #2 (Remarks to the Author):*

*The authors have addressed the reviewers comments extensively*

*Reviewer #3 (Remarks to the Author):*

*Dear authors,*

*I would like to congratulate you on your manuscript. My questions and comments have been fully addressed and*

*implemented in your newly revised manuscript. Therefore I am positive for publication.*
